# DO MLLMS REALLY UNDERSTAND THE CHARTS?

## ABSTRACT

Although Multimodal Large Language Models (MLLMs) have demonstrated increasingly impressive performance in chart understanding, most of them exhibit alarming hallucinations and significant performance degradation when handling non-annotated charts[1]. We argue that current MLLMs rely largely on visual *recognition* rather than visual *reasoning* to interpret the charts, and visual estimation of numerical values is one of the most fundamental capabilities in chart understanding that require complex visual reasoning. To prove this, we introduce ChartVR-Bench, a benchmark meticulously designed to isolate and evaluate visual reasoning ability in chart understanding. Furthermore, we propose ChartVR-3B/7B trained with a novel Visual Reasoning Reinforcement Finetuning (VR-RFT) strategy to strengthen genuine chart visual reasoning abilities. Extensive experiments show that ChartVR achieves superior performance on ChartVRBench, outperforming even powerful proprietary models. Moreover, the visual reasoning skills cultivated by the proposed VR-RFT demonstrate strong generalization, leading to significant performance gains across a diverse suite of public chart understanding benchmarks. The code and dataset will be publicly available upon publication.

## 1 INTRODUCTION

Multimodal Large Language Models (MLLMs) (Bai et al., 2025; Comanici et al., 2025; OpenAI et al., 2024a; Lu et al., 2024) now play a pivotal role in the field of Artificial Intelligence, particularly for understanding complex visual data. These models have demonstrated a remarkable ability to process charts, analyze their content, provide insightful explanations, and achieve competitive performance against existing chart benchmarks (Wang et al., 2024; Masry et al., 2022; Xu et al., 2024b; Masry et al., 2025a; Xia et al., 2025).

Estimating numerical values from charts is a fundamental capability in chart understanding that involves interpreting visual representations to extract or approximate the underlying numbers. The core principle is to understand the mapping between the visual elements (e.g., the position, length, or angle of a mark) on the chart and the data scale it represents. However, when specific numerical annotations are missing from the chart, the propensity of MLLMs to hallucination increases dramatically (Xu et al., 2024b), as exemplified in Figure 1. This leads us to a fundamental question: *Do MLLMs really understand the charts?*

This failure suggests that current MLLMs excel at recognizing about *textual content* within charts but struggle profoundly with reasoning from their underlying *visual geometry*. We argue that it stems from the fundamental reliance of MLLMs on textual *recognition* over genuine visual *reasoning*. To systematically diagnose this core ability, we introduce the Chart Visual Reasoning Benchmark (*ChartVRBench*), which is meticulously designed to isolate numerical value estimation on non-annotated charts, forcing models to move beyond textual recognition. The evaluation reveals that not only open-source MLLMs (Bai et al., 2025; Zhu et al., 2025; Lu et al., 2024) face performance degradation, but even powerful close-source MLLMs, such as GPT-4o (OpenAI et al., 2024a) and Gemini-2.5-Flash (Comanici et al., 2025), also struggle significantly with ChartVRBench.

Moreover, inspired by the success of Reinforcement Learning (RL) in enhancing textual reasoning for mathematics and coding (DeepSeek-AI et al., 2025; OpenAI et al., 2024b; Tan et al., 2025; Huang et al., 2025), we propose ChartVR, a series of MLLMs forged with a novel Visual Reasoning

---

[1]The non-annotated charts are those that require viewers to estimate values using the vertical/horizontal axis scale.

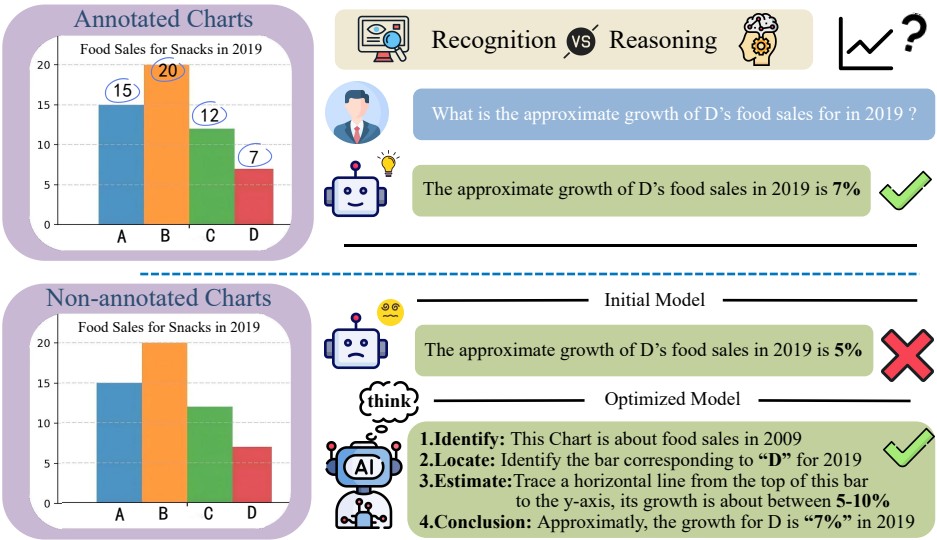

Figure 1: Illustration of the visual reasoning deficit in MLLMs when processing non-annotated charts. A standard model, limited by its underdeveloped visual reasoning capacity, often resorts to guessing and fails. In contrast, our ChartVR executes a deliberate, human-like reasoning chain—identifying the target, locating data based on visual scales, and forming a conclusion—to successfully estimate the value.

Reinforcement Finetuning (VR-RFT) strategy to strengthen genuine chart visual reasoning abilities. The first stage, Visual Reasoning Activation, uses a Chain-of-Thought Supervised Finetuning (CoT-SFT) (Liu et al., 2023) to compel the model to externalize a step-by-step analysis of the chart's visual components. This forces the model to learn an explicit protocol for geometric interpretation, such as locating axes and grounding queries to graphical marks, thereby forming the structural foundation of its visual reasoning capability. Building on this, the second stage, Visual Reasoning Generalization, employs Group Relative Policy Optimization (GRPO) (Shao et al., 2024) to further refine this process. By training on a curated dataset of ambiguous samples where the initial model's judgment is inconsistent, we force it to make finer perceptual discriminations. This training process is guided by a novel continuous accuracy reward function with a quadratic formulation, providing a dense signal directly proportional to the accuracy of the visual estimation. In summary, these stages steer ChartVR to a robust, generalizable visual reasoning capability for charts.

The extensive experiments demonstrate that ChartVR achieves superior performance on ChartVR-Bench, even comparable to powerful proprietary models like Gemini-2.5-Flash (Comanici et al., 2025). More importantly, we demonstrate that the foundational skill cultivated by our method is highly generalizable. ChartVR exhibits significant performance gains across a diverse suite of public, multi-task chart understanding benchmarks (Wang et al., 2024; Xu et al., 2024b; Masry et al., 2025a), proving the effectiveness of our approach in building more rational and reliable MLLMs for chart comprehension.

The main contributions of this work are summarized as follows:

- We introduce *ChartVRBench*, a distinctive benchmark designed to isolate and evaluate genuine visual reasoning capability in chart understanding. It overcomes the limitations of prior work by focusing exclusively on numerical estimation tasks, thus disentangling reasoning from text recognition.

- We propose *ChartVR*, a series of MLLMs with significantly enhanced visual reasoning capabilities for chart understanding. It achieves excellent performance on our challenging ChartVR-Bench, compared with chart-specific and general MLLMs, even surpassing powerful proprietary models like Gemini-2.5-Flash.

- We demonstrate that the visual reasoning ability cultivated by our method is foundational and highly generalizable. *ChartVR* is not confined to the specific numerical estimation task, but

achieves substantial performance gains across a diverse suite of public, multi-task chart understanding benchmarks.

## 2 RELATED WORK

### 2.1 CHART UNDERSTANDING BENCHMARKS

A suite of benchmarks has been developed to evaluate the chart comprehension capabilities of MLLMs. Early benchmarks, such as ChartQA (Masry et al., 2022) and PlotQA (Methani et al., 2020), primarily focused on descriptive tasks. More recently, benchmarks like CharXiv (Wang et al., 2024), ChartQAPro (Masry et al., 2025a), and ChartMuseum (Tang et al., 2025a) have raised the bar by incorporating complex questions and diverse, real-world charts. While these works encompass a wide range of tasks, they often conflate general reasoning with the core challenge of visual interpretation. The most related work to ours is ChartBench (Xu et al., 2024b); while it also focuses on non-annotated charts, it is composed of mostly synthetic data with limited visual diversity. Similarly, recent work by Mukhopadhyay et al. (2024) revealed critical flaws in the consistency and robustness of MLLMs but stopped short of attributing these shortcomings to a fundamental deficit in visual reasoning. We argue this deficit—the core skill of visual reasoning in a chart's geometry, such as numerical value estimation—remains largely untested. Our ChartVRBench is specifically designed to isolate and evaluate this crucial visual reasoning capability.

### 2.2 CHART UNDERSTANDING WITH MLLMS

Many general-purpose MLLMs, such as gpt-4o (OpenAI et al., 2024a), Gemini-2.5 Series (Comanici et al., 2025), and Qwen (Bai et al., 2025), are increasingly applied to chart understanding tasks. In parallel, the development of specialized Chart MLLMs has been rapid, with many models like ChartLlama (Han et al., 2023) and ChartGemma (Masry et al., 2025c). However, their development has largely depended on SFT, a paradigm that, as we argue, tends to cultivate superficial recognition at the expense of genuine reasoning. Recognizing this, a recent wave of models (Chen et al., 2025; Masry et al., 2025b), have incorporated RL to enhance complex, multi-step reasoning. While these RL-based approaches represent a significant step forward, their training objectives often prioritize the final accuracy of text-heavy queries, which can leave the foundational skill of visual grounding underdeveloped. In contrast, our ChartVR is specifically designed to address this fundamental layer. Its RFT framework is meticulously crafted to cultivate the core ability to reason directly from visual geometry, aiming to develop a genuine visual reasoning capability rather than optimizing the textual reasoning that typically follows.

### 2.3 REASONING IN CHART UNDERSTANDING

Reinforcement Learning (RL) has been successfully employed to enhance the reasoning abilities of Large Language Models (LLMs), allowing them to move beyond the static data distributions of SFT (Ouyang et al., 2022). By learning from reward feedback, models have shown significant improvements in complex domains like mathematics and coding (DeepSeek-AI et al., 2025; Shao et al., 2024). Inspired by this success, several works have begun to apply similar RL-based paradigms to MLLMs (Feng et al., 2025; Tan et al., 2025; Huang et al., 2025), activating their visual reasoning on tasks like visual counting and spatial transformation. Building on these advancements, our work adapts this powerful paradigm to the specialized domain of chart understanding.

## 3 CHARTVRBENCH

We introduce Chart Visual Reasoning Benchmark (ChartVRBench), a comprehensive, multi-domain, and reasoning-centric benchmark designed to rigorously assess the visual interpretation capabilities of MLLMs on charts that lack explicit numerical annotations. Engineered to move beyond simple OCR-dependent tasks, the benchmark comprises a total of 2,453 question-answer pairs. It features a majority (2,101 pairs) of synthetically generated charts to ensure controlled complexity and a significant portion (352 pairs) sourced from real-world examples to guarantee practical relevance.

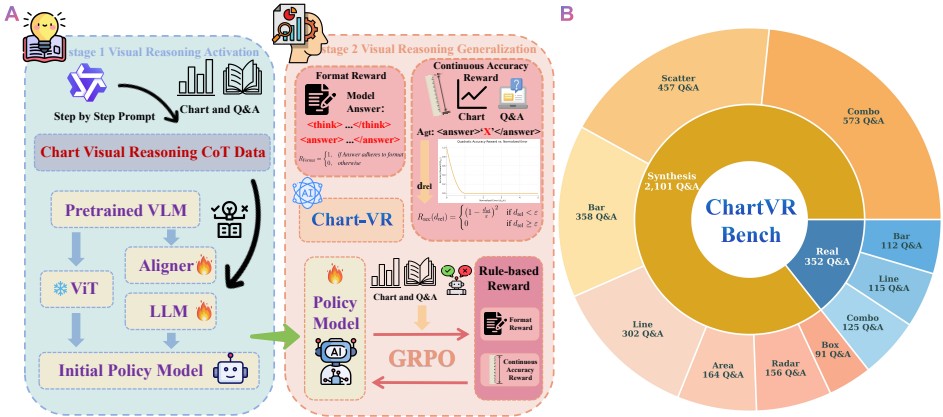

Figure 2: The training paradigm of ChartVR and the data distribution of ChartVRBench. A: ChartVR leverages a two-stage RFT strategy. Stage 1 activates the model's reasoning abilities via SFT on CoT data, while Stage 2 uses GRPO with a multi-component reward system to reinforce correct chart understanding. B: The composition of ChartVRBench, detailing the distribution of seven chart types across both synthetic and real data sources.

Table 1: Comparison between ChartVRBench and existing representative chart QA benchmarks. Symbols: ✓ Fully Supported / High Quality; △ Partially Supported / Mixed; × Not Supported / Low Quality.

| Feature | EvoChart Huang et al. (2024) | ChartBench Xu et al. (2024b) | CharXiv Wang et al. (2024) | ChartQAPro Masry et al. (2025a) | ChartMuseum Tang et al. (2025a) | Ours |
|---|---|---|---|---|---|---|
| Real-World Charts | ✓ | × | ✓ | ✓ | ✓ | ✓ |
| Broad Topic Coverage | × | × | △ | △ | △ | ✓ |
| Non-Annotated Charts | × | ✓ | × | × | × | ✓ |
| Isolates Visual Reasoning | × | △ | × | × | ✓ | ✓ |

The benchmark provides extensive coverage across seven primary chart types, including bar, line, scatter, and combo charts, with a detailed breakdown of the data distribution shown in Figure 2. This structural diversity is complemented by thematic breadth, with data spanning 38 distinct topics, including finance, healthcare, and technology. This dual emphasis on structural and thematic variety ensures a rigorous evaluation, mitigating the risk of models overfitting to specific chart formats or familiar domains.

While existing benchmarks have significantly advanced the field, they predominantly focus on general high-level Question Answering (QA), where visual reasoning is often conflated with textual extraction (OCR) and logical reasoning. ChartVRBench fills a critical gap by strictly isolating the visual reasoning capability of numerical value estimation on non-annotated charts, preventing models from relying on text recognition shortcuts.To clearly demonstrate how our benchmark compares to contemporary works, we present a feature-wise comparison in Table 1.

### 3.1 DATA CURATION

**Synthetic Chart Generation.** Our synthetic chart generation process is partially adapted from the Code-as-Intermediary Translation (CIT) methodology proposed by He et al. (2024), where executable plotting code serves as the ground truth for each chart. The process begins with a curated set of seed scripts, which are then programmatically diversified using Self-Instruct (Wang et al., 2023) and Evol-Instruct (Xu et al., 2024a) techniques to generate a vast library of visually complex charts. A critical constraint is the deliberate omission of numerical labels on data points, ensuring that every chart necessitates visual estimation. To maximize yield, a self-repair mechanism leverages an LLM to debug and correct any code that fails during execution. Following an automated visual fidelity check by a MLLM, the entire collection of synthesized data underwent a final, rigorous human review. This manual verification step served to confirm the high quality of the chart images and the accuracy of their corresponding question-answer pairs. This code-centric approach, combined

with multiple stages of validation, provides an unimpeachable ground truth, allowing us to generate verifiably correct Q&A pairs.

**Real Chart Collection.** To anchor our benchmark in real-world applications, we sourced charts from reputable data repositories such as Statista and Our World in Data. Each chart was manually vetted by human annotators to meet strict criteria: high visual quality, data integrity, and a complete absence of explicit numerical annotations. Following selection, an MLLM was used to generate candidate question-answer pairs for each chart. Every MLLM-generated pair then underwent a final round of human verification and refinement to guarantee the accuracy and relevance of both the question and its ground-truth answer.

## 3.2 EVALUATION PROTOCOL

Standard exact-match accuracy is ill-suited for value estimation from non-annotated charts, as it fails to account for the slight perceptual ambiguity inherent in the task, even for human observers. To address this, we employ a relaxed accuracy metric, which judges a prediction correct if its relative error from the ground-truth value falls within a tolerance threshold, denoted as $\tau$. To align this threshold with human performance, we conducted an empirical study and found that human estimations consistently fall within a 2% error margin. Accordingly, we empirically set $\tau = 0.02$.

Formally, a model's predicted value, $A_{pred}$, is deemed correct if and only if it satisfies the following condition relative to the ground truth, $A_{gt}$:

$$A_{pred} \in [(1 - \tau) \times A_{gt}, (1 + \tau) \times A_{gt}]$$

This protocol ensures that our evaluation is both rigorous and fairly aligned with human-level interpretive capabilities, rewarding models for precise visual reasoning rather than penalizing them for minor, human-like estimation variance.

## 4 CHARTVR

We propose ChartVR, a series of MLLMs designed to perform visual reasoning for better visual understanding on non-annotated charts. We formally define this task as follows: given a chart image $I$, and a corresponding textual question $Q$, the goal is to derive a numerical answer $A$ with a reasoning procedure $R$. This process can be represented as a mapping function $\mathcal{F}$:

$$\mathcal{F} : (I, Q) \to (R, A)$$

where $I$ is the chart image, $Q$ is the question in text, $R$ is the step-wise reasoning procedure in text, and $A \in \mathbb{R}$ is the numerical answer. The fundamental challenge lies in interpreting non-annotated charts, which requires the model to reason about geometric structures (e.g., axes, scales, positions) to infer values, rather than simply extracting them via text recognition.

To address this challenge, we propose a novel two-stage Reinforcement Finetuning (RFT) framework. This approach is designed to first instill a robust, human-like reasoning framework and then meticulously refine the model's numerical e precision. As illustrated in Figure 2, the RFT pipeline consists of two sequential stages: (1) Visual Reasoning Activation, which uses supervised fine-tuning to teach the model the structure of reasoning, followed by (2) Visual Reasoning Generalization, which uses reinforcement learning to improve the accuracy and generalizability.

### 4.1 STAGE 1: VISUAL REASONING ACTIVATION

The initial stage of our pipeline aims to establish a foundational reasoning paradigm. Instead of having the model directly guess an answer, we teach it to adopt a structured, step-by-step thought process that mirrors human analysis. To achieve this, we fine-tune our base model on a high-quality dataset of 43k samples generated by distilling detailed Chain-of-Thought (CoT) processes from an advanced MLLM (see Appendix B.1 for details). This CoT-SFT process systematically teaches the model to move beyond direct answer prediction and instead adopt a structured analytical approach: first identifying and utilizing critical chart components—such as axes, scales, and legends—and then using them to derive a final answer.

Formally, we employ SFT on this dataset. Each data instance is a tuple $(x, q, r, a)$, where $x$ is the chart image, $q$ is the question, $r$ is the intermediate reasoning chain, and $a$ is the final answer. The training objective is to minimize the negative log-likelihood of the model generating the complete sequence $y$ (the concatenation of $r$ and $a$) given the image $x$ and question $q$:

$$\mathcal{L}_{\text{SFT}} = -\mathbb{E}_{(x,q,r,a)\sim\mathcal{D}} \sum_{t=1}^{|y|} \log \pi_\theta(y_t|x, q, y_{<t}) \tag{1}$$

where $\mathcal{D}$ is our CoT dataset and $\pi_\theta$ is the policy of the model with parameters $\theta$. The resulting fine-tuned model, denoted as $\pi_{\text{SFT}}$, learns a robust template for visual reasoning and serves as the starting point for the next stage.

## 4.2 STAGE 2: VISUAL REASONING GENERALIZATION

Building on the visual reasoning foundation from Stage 1, the second stage focuses on enhancing the model's precision and reliability for the numerical estimation task. For this, we use a smaller, high-signal dataset of 3.4k samples curated to target the model's specific weaknesses. These samples are identified by selecting problems where the SFT-tuned model exhibits "stochastic correctness"—that is, problems it can solve but not consistently (see Appendix B.3 for details). By training on these borderline cases with higher-resolution images, we force the model to refine its visual interpretation skills.

We employ GRPO (Shao et al., 2024), an efficient and scalable reinforcement learning algorithm, to fine-tune the policy model $\pi_{\text{SFT}}$. Unlike traditional algorithms like PPO (Schulman et al., 2017), GRPO forgoes a computationally expensive value network and instead calculates relative advantages by comparing rewards within a group of sampled responses. For each input $(x, q)$, we sample a group of $G$ candidate answers $\{a_1, a_2, \ldots, a_G\}$ from the current policy $\pi_\beta$. Each answer $a_i$ receives a reward $R(a_i)$, and these rewards are used to compute a normalized relative advantage $A_i$ for each sample:

$$A_i = \frac{r_i - \text{mean}\{r_1, \ldots, r_G\}}{\text{std}\{r_1, \ldots, r_G\}} \tag{2}$$

The policy is then updated to increase the probability of actions with positive advantages, while a KL-divergence penalty against the reference model $\pi_{\text{SFT}}$ ensures stable training.

## 4.3 REWARD FUNCTION DESIGN

The effectiveness of our RL stage hinges on a well-designed reward function. Our function $R(a_i)$ is a composite of two components, targeting both response structure and numerical accuracy:

$$R(a_i) = R_{\text{format}}(a_i) + R_{\text{acc}}(a_i) \tag{3}$$

**Format Reward.** To encourage interpretable and well-structured outputs, we provide a binary format reward, $R_{\text{format}}$. The model receives a reward of 1 if its response strictly adheres to our predefined template, where reasoning is enclosed in `<think></think>` tags and final answer in `<answer></answer>` tags, and 0 otherwise.

**Continuous Accuracy Reward.** To overcome the sparse signal from a simple correct/incorrect binary reward, we introduce a continuous accuracy reward, $R_{\text{acc}}$. This reward provides a fine-grained signal that recognizes "nearly correct" answers. For a predicted answer $A_{\text{pred}}$ and a non-zero ground truth $A_{\text{gt}}$, we first calculate the relative error:

$$d_{\text{rel}} = \frac{|A_{\text{pred}} - A_{\text{gt}}|}{|A_{\text{gt}}|} \tag{4}$$

Then, we define the reward using a piecewise quadratic function that smoothly decays from 1 to 0:

$$R_{\text{acc}}(d_{\text{rel}}) = \begin{cases} \left(1 - \frac{d_{\text{rel}}}{\tau}\right)^2 & \text{if } d_{\text{rel}} < \tau \\ 0 & \text{if } d_{\text{rel}} \geq \tau \end{cases} \tag{5}$$

Table 2: Comparison of ChartVR with representative MLLMs on the proposed ChartVRBench. The best and second-best scores in each column are highlighted using bold and underline formatting, respectively.

| Methods | Synthetic Charts | | | | | | | Real Charts | | | Overall |
| --- | --- | --- | --- | --- | --- | --- | --- | --- | --- | --- | --- |
| | Box | Area | Radar | Scatter | Bar | Line | Combo | Bar | Line | Combo | |
| Human Evaluation | 94.51 | 43.29 | 88.46 | 91.24 | 96.65 | 97.68 | 90.92 | 84.82 | 84.35 | 65.60 | 87.57 |
| *Open-source Models* | | | | | | | | | | | |
| InternVL3-2B (Zhu et al., 2025) | 25.27 | 8.54 | 9.62 | 25.16 | 51.68 | 43.05 | 34.55 | 43.75 | 34.78 | 32.00 | 32.98 |
| Qwen2.5-vl-3B (Bai et al., 2025) | 46.15 | 14.02 | 17.95 | 51.42 | 72.91 | 81.13 | 62.83 | 66.96 | 54.78 | 45.60 | 56.62 |
| Ovis1.6-llama3.2-3B (Lu et al., 2024) | 12.09 | 3.66 | 7.69 | 14.66 | 13.13 | 10.93 | 11.69 | 8.04 | 16.52 | 12.00 | 11.66 |
| Gemma-3-4B (Team et al., 2025) | 9.89 | 3.05 | 12.82 | 11.82 | 9.22 | 12.25 | 6.81 | 8.04 | 12.17 | 7.20 | 9.34 |
| Qwen2.5-vl-7B (Bai et al., 2025) | 70.33 | 21.34 | 19.23 | 61.93 | 73.74 | 85.43 | 68.41 | 49.11 | 56.52 | 48.00 | 61.39 |
| InternVL3-8B (Zhu et al., 2025) | 36.73 | 12.80 | 12.18 | 39.17 | 43.58 | 47.68 | 36.82 | 38.39 | 46.09 | 34.40 | 36.73 |
| *Close-source Models* | | | | | | | | | | | |
| GPT-4o (OpenAI et al., 2024a) | 28.57 | 12.20 | 11.54 | 25.61 | 21.23 | 27.15 | 18.15 | 13.39 | 26.96 | 18.40 | 20.87 |
| Gemini-2.5-Flash (Comanici et al., 2025) | 68.13 | 25.61 | 7.69 | 61.93 | 72.07 | 75.17 | 55.85 | 49.11 | 52.17 | 39.20 | 55.77 |
| *Chart-Specific Models* | | | | | | | | | | | |
| ChartGemma-3B (Masry et al., 2025c) | 10.99 | 10.98 | 7.05 | 21.44 | 42.74 | 32.78 | 24.43 | 37.50 | 42.61 | 28.80 | 26.74 |
| TinyChart-3B (Zhang et al., 2024) | 13.19 | 7.93 | 7.69 | 25.16 | 56.15 | 54.30 | 36.65 | 57.14 | 40.87 | 34.40 | 35.83 |
| ChartInstruct-7B (Masry et al., 2024) | 10.99 | 1.22 | 4.49 | 16.63 | 35.47 | 16.56 | 18.85 | 51.79 | 45.22 | 18.40 | 20.91 |
| ChartVLM-7.3B (Xia et al., 2025) | 9.89 | 10.37 | 7.69 | 10.28 | 70.39 | 45.36 | 32.81 | 50.00 | 54.78 | 35.20 | 33.63 |
| ChartLlama-13B (Han et al., 2023) | 10.99 | 1.83 | 5.77 | 5.25 | 3.35 | 4.30 | 4.01 | 8.04 | 9.57 | 7.20 | 5.01 |
| Bespoke-MiniChart-7B (Tang et al., 2025b) | 72.53 | 26.83 | 25.00 | 66.74 | 86.87 | 89.40 | 75.04 | **69.64** | **62.61** | 45.60 | 68.16 |
| Chart-R1 (7B) (Chen et al., 2025) | **83.52** | 26.22 | 25.64 | 66.83 | 85.47 | 82.45 | 68.94 | 61.61 | 59.13 | 44.80 | 65.72 |
| **ChartVR-3B (Ours)** | 63.74 | 24.39 | 19.87 | 57.33 | 82.68 | 87.75 | 69.46 | 58.93 | 57.39 | 45.60 | 62.74 |
| **ChartVR-7B (Ours)** | **83.52** | **37.20** | **27.56** | **70.90** | **88.27** | **92.05** | **78.53** | **69.64** | **62.61** | **58.40** | **72.20** |

We empirically set $\tau = 0.02$ based on a human-calibrated tolerance threshold. For the specific case where the ground truth $A_{gt}$ is zero, because it is difficult to quantize the relative deviations, the accuracy reward falls back to exact match, assigning a value of 1 when $A_{gt} = A_{pred}$ and 0 otherwise.

We employ the quadratic formulation for two critical reasons. First, this design provides a clear, bounded, and intuitive reward range. It yields a reward of 1 for a perfect answer ($d_{rel} = 0$) and smoothly decay to 0 as the relative error hits the 2% tolerance boundary. Second, the quadratic shape creates a desirable non-linear decay. It has a gentle slope for subtle errors, granting substantial partial credit for close answers, while the penalty accelerates as the error approaches the tolerance threshold. This behavior encourages the model to make fine-grained improvements when it is already close to the correct answer, while strongly penalizing larger, unacceptable deviations.

## 5 EXPERIMENTS

### 5.1 EXPERIMENTAL SETUPS

**Implementation Details.** The implementation was built upon the ModelScope SWIFT framework (Zhao et al., 2025). We initialize our ChartVR models using the open-source Qwen2.5-VL series (Bai et al., 2025) as a foundation. For inference, all models and benchmarks follow their provided settings where available, with results obtained from a single forward pass using a fixed random seed of 42 to ensure reproducibility. Additional details are available in Appendix C.

**Main Evaluation on ChartVRBench.** Our primary evaluation is conducted on the proposed ChartVRBench to assess genuine visual reasoning capabilities and establish the superiority of our ChartVR model. On this benchmark, we compare our model against a comprehensive suite of baselines organized into three categories: open-source MLLMs, powerful close-source MLLMs and prominent chart-specific models.

**Generalization Study on Public Benchmarks.** To evaluate the transferability of the skills learned via our RFT framework, we conduct a generalization study. Specifically, we benchmark our ChartVR model against the representative chart-specific models—ChartGemma (Team et al., 2025), TinyChart (Zhang et al., 2024), ChartInstruct (Masry et al., 2024), ChartVLM (Xia et al., 2025),

Table 3: Performance of ChartVR compared to other chart-specific models on various public chart understanding benchmarks. All results are reproduced by the authors. Qwen2.5-VL baselines are listed below their corresponding ChartVR models with an arrow indicator. The best and second-best scores in each column are highlighted using bold and underline formatting, respectively.

| Models | ChartVRBench | CharXiv (R) | ChartBench | ChartQAPro |
|---|---|---|---|---|
| ChartGemma-3B (Masry et al., 2025c) | 26.74 | 12.50 | - | 6.84 |
| TinyChart-3B (Zhang et al., 2024) | 35.83 | 8.30 | - | 13.25 |
| ChartInstruct-7B (Masry et al., 2024) | 20.91 | 8.80 | - | 4.88 |
| ChartVLM-7.3B (Xia et al., 2025) | 33.63 | - | 12.06 | - |
| ChartLlama-13B (Han et al., 2023) | 5.01 | 14.20 | 21.30 | - |
| Bespoke-MiniChart-7B (Tang et al., 2025b) | 68.16 | 41.40 | 44.19 | 34.74 |
| Chart-R1 (7B) (Chen et al., 2025) | 65.80 | **50.00** | 15.71 | 31.81 |
| **ChartVR-3B (Ours)** | 62.74 | 33.40 | 26.35 | 28.03 |
| ↪ *Qwen2.5-VL-3B (Bai et al., 2025)* | 56.62 | 30.60 | 21.30 | 24.51 |
| **ChartVR-7B (Ours)** | **72.20** | 43.40 | **45.34** | **41.79** |
| ↪ *Qwen2.5-VL-7B (Bai et al., 2025)* | 61.39 | 39.50 | 35.35 | 37.10 |

and ChartLlama (Han et al., 2023)—on a diverse set of public benchmarks. This suite includes the real-world datasets CharXiv (Wang et al., 2024) and ChartQAPro (Masry et al., 2025a), as well as the synthetic benchmark ChartBench (Xu et al., 2024b). This allows us to verify that our training methodology imparts a foundational reasoning ability that generalizes effectively to a variety of chart understanding tasks.

## 5.2 EXPERIMENTAL RESULTS

**Performance on ChartVRBench.** The results, presented in Table 2, underscore the significant challenge that ChartVRBench poses to a wide range of MLLMs. The generally low scores across all categories—including powerful proprietary models like GPT-4o (20.87%) and Gemini-2.5-Flash (55.77%)—reveal a critical and widespread weakness in genuine visual reasoning. This difficulty stems from our benchmark's design, which forces models to infer values from graphical geometry (e.g., axes and scales) rather than relying on OCR-based shortcuts common in other benchmarks. The particularly low score of GPT-4o has been double-checked and manually validated, which precisely indicates that many MLLMs lack the genuine visual reasoning ability, as we argue.

Our proposed model, ChartVR, demonstrates a clear superiority in this demanding task. ChartVR-7B achieves an overall score of 72.20%, outperforming all other models, including the best open-source baseline, Qwen2.5-vl-7B (61.39%), and the strongest proprietary model, Gemini-2.5-Flash. Notably, even our smaller ChartVR-3B model (62.74%) surpasses most other models, highlighting the effectiveness of our training methodology. The performance is particularly strong on complex chart types like Real Combo charts, where ChartVR-7B (58.40%) dramatically outperforms the other models.

**Generalization on Public Benchmarks.** As detailed in Table 3, ChartVR-7B exhibits exceptional generalization, achieving competitive results across the board. Specifically, on the reasoning-focused portion of CharXiv, our model achieves an improvement of 3.9% over the base model Qwen2.5-VL-7B. This success stems from our model's core visual reasoning capability, which contrasts with other chart-specific models that rely on SFT and often fail to develop a generalizable reasoning capability.

## 5.3 ABLATION STUDY

**Effectiveness of the RFT Training Paradigm.** To systematically validate our training strategies, we conducted an comparative study comparing three paradigms: CoT-SFT, GRPO applied directly to the base model, and our full RFT framework. The results are summerized in Table 4. For ChartVR-7B, the base model scores 61.39%. Using CoT data as a 'Visual Reasoning Activation' merely compels the model to adopt a visual reasoning pattern without imparting the underlying ability. Consequently, this mismatch leads to a performance degradation rather than an improvement.

Table 4: Ablation study of training strategies for ChartVR-3B and ChartVR-7B models. The best and second-best scores are highlighted.

Table 5: Ablation study of reward function components for Qwen2.5-VL-7B.

| Training Paradigm | ChartVR-3B | | | ChartVR-7B | | |
|---|---|---|---|---|---|---|
| | Synthetic | Real | Overall | Synthetic | Real | Overall |
| Zero-Shot | 56.87 | 54.78 | 56.52 | 67.66 | 51.16 | 61.39 |
| +CoT-SFT | 36.55 | 34.86 | 40.77 | 55.45 | 45.74 | 54.06 |
| +GRPO | 56.59 | 56.82 | 56.62 | 66.44 | 54.83 | 64.78 |
| +RFT | 64.26 | 53.69 | 62.74 | 73.68 | 63.35 | 72.20 |

| Reward Component | Score |
|---|---|
| Format | 61.84 |
| Cont. Acc | 67.88 |
| Acc + Format | 70.28 |
| **Cont. Acc + Format** | **72.20** |

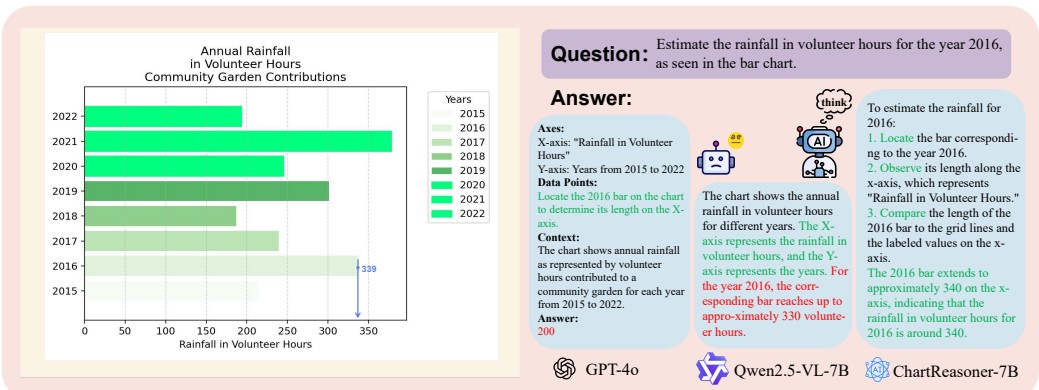

Figure 3: Qualitative analysis on ChartVRBench. GPT-4o and Qwen2.5-VL-7B exhibits hallucination in the answer and reasoning procedure, respectively. In contrast, our ChartVR-7B is able to produce a coherent and correct step-by-step reasoning process, leading to an accurate answer.

In contrast, the full RFT framework, which synergistically combines SFT with RL, achieves the most significant performance gain, reaching 72.20%. This demonstrates that our complete VR-RFT framework genuinely enhances the model's visual reasoning ability, leading to its consistent and substantial outperformance over all other configurations.

**Impact of the Reward Function Design.** We conducted an ablation study to isolate the contribution of each component in our reward function, with results presented in Table 5. The findings highlight a strong synergistic effect between enforcing a correct output structure and rewarding numerical precision. A model trained with only the Format reward achieves a score of 61.84, whereas combining it with our proposed Continuous Accuracy (Cont. Acc) reward boosts the score significantly to 72.20, indicating both components are crucial. Furthermore, the study validates the superiority of our continuous reward design over a standard binary alternative. A model using a simple binary accuracy reward (Acc + Format), which provides a sparse correct/incorrect signal, is clearly outperformed by our model using the continuous reward (Cont. Acc + Format). This demonstrates the effectiveness of our quadratic reward function, which provides a dense and informative learning gradient. By rewarding "nearly correct" answers, it encourages the fine-grained improvements in visual estimation necessary for achieving higher precision.

## 5.4 CASE STUDY

In Figure 3, powerful models like GPT-4o and Qwen2.5-VL-7B either misinterpret the query or resort to factual hallucination. In contrast, our ChartVR, sculpted by the RFT strategy, demonstrates a flawless, step-by-step reasoning process. This case study provides a compelling visual proof of our quantitative findings: ChartVR enhanced by the RFT strategy is transformed from a system prone to errors and hallucination into a reliable and structured visual reasoner that moves beyond merely optimizing for the answer.

## 6 Conclusion

In this paper, we investigate a compelling yet significant question: "Do MLLMs really understand the charts?" By establishing the ChartVRBench, we extensively evaluated open-source, close-source, and chart-specific MLLMs. The results shows a significant degradation in the performance of these models, and, through chain-of-thought reasoning, revealed their inability to estimate numerical values through visual reasoning, similar to human behavior. To address this issue, we propose ChartVR, which enhances the visual reasoning ability of MLLMs via an RFT strategy. This strategy first activates reasoning capabilities through SFT, and then generalizes reasoning abilities through RL. Extensive experiments conducted on the proposed ChartVRBench and public chart reasoning datasets demonstrate the effectiveness of ChartVR. This work paves the way for empowering MLLMs to really understand the charts in a human-like manner.

### Ethics Statement

This research adheres to the ICLR Code of Ethics. The *ChartVRBench* dataset was constructed with ethical considerations as a priority, with real-world chart data sourced exclusively from publicly accessible platforms (Statista and Our World in Data) in strict adherence to their terms of service. Human participation was limited to the ethical recruitment and fair compensation of annotators for the curation of question-answer pairs and for a small-scale study to calibrate our human-aligned evaluation metric. The purpose of all tasks was clearly communicated. We believe our work, which aims to foster more reliable and less hallucinatory AI systems by promoting genuine visual reasoning, does not raise any major negative societal concerns.

### Reproducibility Statement

To ensure the full reproducibility of our findings, all code, data, and trained models will be made publicly available upon publication. The source code for our novel Reinforcement Finetuning (RFT) framework, including implementations for both the SFT and GRPO stages, along with all evaluation scripts, will be released. The complete *ChartVRBench* dataset, including data generation scripts for the synthetic portion, will also be made available. Detailed descriptions of the data curation process, training hyperparameters, and evaluation protocols are provided in Appendices A, B, and C, respectively. Finally, we will release the final weights for our trained ChartVR-3B and ChartVR-7B models to facilitate future research.

### LLM Usage Statement

LLMs were utilized as general-purpose assistants throughout this research project. In the research phase, LLMs served as a tool to accelerate our workflow by generating professional experiment code, assisting with bug fixing, and conducting deep research to help discover novel ideas and related works. During the preparation of this manuscript, LLM was also used as a writing and editing assistant to polish prose for clarity, improve the narrative flow of sections, and format complex LaTeX tables. All scientific claims, data analysis, and final conclusions were determined by the human authors, who have reviewed all generated and modified content to ensure its correctness and take full responsibility for the scientific integrity of this paper.

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

APPENDIX

- Sec. A provides details of the proposed benchmark, *ChartVRBench*, including the problem definition, data sources, topics, and chart types.

- Sec. B describes the training strategy of the *ChartVR*, detailing the construction of the SFT and RL datasets.

- Sec. C presents further details on the evaluation and inference.

- Sec. D interpret the human performance in ChartVRBench.

# A    CHARTVRBENCH BENCHMARK DETAILS

## A.1    PROBLEM DEFINITION

The primary task addressed by our benchmark, *ChartVRBench*, is numerical value estimation on non-annotated charts. Formally, given a chart image $C$ and a query $Q$ that specifies a target data point within the chart, the goal is to produce a numerical answer $A$ that accurately estimates the value of that data point. Crucially, the chart image $C$ is non-annotated, meaning that the numerical values corresponding to graphical elements (e.g., the height of a bar, a point on a line) are not present as explicit text labels.

This task is fundamentally a visual reasoning problem, rather than a simple recognition or textual reasoning task. To arrive at the correct answer $A$, a model cannot only rely on Optical Character Recognition (OCR). Instead, it must perform a multi-step cognitive process grounded in the visual geometry of the chart:

1. Semantic Understanding & Grounding: The model must first parse the query $Q$ and correctly associate the textual description with the corresponding graphical elements in the chart image $C$ (e.g., a specific bar, a specific colored line).

2. Structural and Scale Interpretation: The model must identify and interpret the chart's structural components, particularly the relevant axes (e.g., the y-axis) and their numerical scales, including the range and the value represented by tick marks and grid lines.

3. Spatial and Proportional Reasoning: Finally, the model must perform spatial reasoning by comparing the target graphical element's dimension (e.g., its height or position) against the interpreted scale of the axis. This often requires proportional estimation or interpolation between labeled tick marks to infer the final numerical value.

By designing a task that necessitates this entire reasoning chain, we directly evaluate a model's ability to not just *recognize* a chart, but to truly *understand* its underlying quantitative information.

## A.2    DATA TOPICS AND CHART EXAMPLES

To ensure the breadth relevance of *ChartVRBench*, the synthetic data generation process samples from a diverse set of 38 distinct topics, as shown in Table 6 guaranteeing that the charts cover a variety of contexts and narratives. Furthermore, to robustly evaluate a model's visual reasoning capabilities across different graphical representations, ChartVRBench incorporates seven primary chart types. Figure 5 provides a representative example for each of these types, showcasing the visual diversity and complexity present in our benchmark.

## A.3    REAL-WORLD CHART COLLECTION

To ensure *ChartVRBench* reflects the challenges of real-world applications, we curated a substantial collection of charts from public online sources. This section details our three-stage process: data sourcing, a rigorous manual filtering protocol, and a hybrid human-AI pipeline for generating high-quality question-answer pairs. Figure 6 showcases several examples of the final curated real-world charts from our collection.

Table 6: The 38 topics covered in the ChartVRBench dataset.

| Category | Category |
| --- | --- |
| Agriculture and Food Production | Human Resources and Employee Management |
| Architecture and Building | Language and Communication |
| Artificial Intelligence and Robotics | Law and Legal Affairs |
| Art and Design | Literature and Writing |
| Astronomy and Space | Manufacturing and Production |
| Biology and Life Sciences | Marketing and Advertising |
| Books and Publishing | Mathematics and Statistics |
| Business and Finance | Music and Performance |
| Computer Science and Information Technology | Physics and Chemistry |
| Education and Academics | Real Estate and Housing Market |
| Energy and Utilities | Religion and Spirituality |
| Environment and Sustainability | Retail and E-commerce |
| Fashion and Style | Science and Engineering |
| Film and Cinema | Social Media and the Web |
| Food and Beverage Industry | Social Sciences and Humanities |
| Futurism and Innovation | Society and Community |
| Government and Public Policy | Sports and Entertainment |
| Healthcare and Health | Transportation and Logistics |
| History and Culture | Travel and Exploration |

### A.3.1 DATA SOURCING

**Statista.** A significant portion of the real-world charts was sourced from Statista[2], a leading global platform specializing in market and consumer data. Statista provides professional, data-driven visualizations for a wide array of industries, covering topics from economic indicators and market forecasts to technology trends and consumer behavior.

**Our World in Data.** The second major source was Our World in Data[3], a renowned scientific online publication based at the University of Oxford. Its mission is to make data and research on the world's most significant challenges, such as global health, economic development, and environmental change, accessible and understandable through complex and data-rich visualizations.

In addition to these two primary repositories, the collection was supplemented by charts from various other miscellaneous public reports and online publications. All data collection was conducted in strict adherence to the copyright policies, terms of service, and licensing agreements of each source to ensure full ethical compliance.

### A.3.2 CURATION PROTOCOL

Once a large pool of charts was collected, each candidate chart underwent a meticulous, two-step curation process performed by our recruited team of human annotators.

**Step 1: Manual Filtering and Vetting.** Each chart was manually vetted against stringent criteria for inclusion in *ChartVRBench*. A chart was accepted only if it satisfied all of the following conditions, otherwise it was discarded:

1. High Visual Quality: The image must be of sufficient resolution, clear, and free of significant compression artifacts or other visual noise that could impede interpretation.

2. Data Integrity: The chart must be coherent and visually consistent, with clearly defined axes, legends, and graphical elements.

3. Absence of Annotations: We exclusively select charts where numerical values are not explicitly printed on the graphical elements (e.g., no numbers on top of bars). This constraint is funda-

---

[2]https://www.statista.com/
[3]https://ourworldindata.org/

mental to our benchmark, as it forces a model to perform genuine visual reasoning rather than relying on OCR shortcuts.

**Step 2: Question-Answer Pair Generation.** Once a chart was approved, we employed a two-stage, human-in-the-loop process to generate its corresponding question-answer pair:

1. MLLM-based Candidate Generation: We first use a capable MLLM to generate an initial set of candidate question-answer pairs for each chart, prompting it to ask a specific numerical estimation question.

2. Human Verification and Refinement: Every MLLM-generated pair is then subjected to rigorous human review. Annotators verify the question's clarity and relevance, and then carefully perform the visual estimation themselves to validate the answer's accuracy. If necessary, they refine the question's phrasing or correct the ground-truth answer to ensure the final Q&A pair is unambiguous and factually correct.

# B DETAILS OF DATASETS AND TRAINING

## B.1 DATA CONSTRUCTION FOR SFT

The dataset used for the Supervised Fine-Tuning (SFT) "cold start" phase is meticulously constructed through a **knowledge distillation** process. The goal is to generate high-quality Chain-of-Thought (CoT) data that can effectively activate the reasoning paradigm of our base models.

While we utilize the same underlying generation pipeline (rendering engine and topic distribution) to ensure domain consistency, the specific chart instances and question-answer pairs in the SFT set are distinct from those in the benchmark. Crucially, to maintain strict train-test separation, this SFT dataset is generated as a completely independent batch from the ChartVRBench evaluation set.

This process leverages a powerful teacher model (Qwen2.5-VL-32B-Instruct) to generate reasoning traces for this training-specific corpus. The data construction pipeline involves several key stages to ensure the quality and validity of the final CoT samples:

1. **Prompting for Chain-of-Thought Generation:** For each generated training instance, consisting of Python plotting code (`C`) and a question (`Q`), we employ the teacher LLM to generate a step-by-step reasoning process. The core of this process involves a carefully constructed textual prompt that integrates the Python code (`C`) and the question (`Q`). This prompt is designed to force the model to articulate a logical pathway from the code and question to the correct result. The model is required to structure its output using specific tags, separating the reasoning steps (`<think>...</think>`) from the final answer (`<answer>...</answer>`).

2. **Validation and Filtering:** Each generated CoT sample undergoes a rigorous, multi-step validation process to filter out low-quality or incorrect reasoning:
   - *Structural Check:* The generated text is first parsed to ensure that both the reasoning and answer tags are present. Samples with missing tags are discarded.
   - *Answer Verification:* The final answer extracted from the `<answer>` tag is programmatically compared against the ground-truth answer derived from the code. We employ a robust evaluation function that checks for both exact string matches and numerical equivalence within a tolerance threshold to ensure correctness.
   - *Leakage Detection:* The generated reasoning trace within the `<think>` tags is scanned for any mention of the "original answer." This crucial step prevents the model from "cheating" by simply copying the ground-truth answer into its reasoning, ensuring that the generated thought process is genuine.

## B.2 RL ALGORITHM SELECTION

We selected GRPO to fine-tune our multimodal model for enhanced chart visual reasoning, primarily due to its superior efficiency and its alignment with our reward structure. Compared to Proximal Policy Optimization (PPO), GRPO significantly reduces computational and memory overhead. GRPO eliminates the need for a separate value model—which is typically as large as the policy model—by

estimating the baseline directly from the scores of multiple sampled outputs. This efficiency is critical given the large scale of our models (e.g., 7B parameters), making GRPO a practical solution under limited hardware resources.

Furthermore, while Direct Preference Optimization (DPO) offers an efficient alternative to traditional RLHF, it is fundamentally designed for binary preference datasets (i.e., chosen vs. rejected responses). Our task, however, benefits from a more granular, continuous reward signal that reflects the degree of correctness in quantitative analysis. GRPO is adept at directly optimizing for such programmatic, scalar rewards, allowing the model to learn from fine-grained feedback. This makes it better suited for improving the visual reasoning in chart than a preference-based method like DPO.

### B.3 DATASET CONSTRUCTION FOR GRPO

The curation process, inspired by the principles of rejection sampling and active learning, involves a multi-round, varied-prompting inference pipeline designed to probe the model's knowledge boundaries. The goal of this pipeline is to construct a specialized, high-signal dataset by isolating ambiguous cases that the SFT-tuned model can sometimes solve but not consistently. This strategy focuses the training process on the most informative examples where the model is most uncertain, rather than wasting computational resources on problems that are already mastered (always correct) or are currently too difficult (always incorrect).

Our pipeline involves the following systematic steps:

1. **Initial Dataset Curation:** We begin by constructing an initial, high-quality dataset for GRPO. This dataset is synthesized following the method introduced earlier. We selected only those instances that achieved a score of either 4 or 5, ensuring a strong baseline of correct.

2. **Multi-Round Inference:** Initially, we run inference multiple times on the initial GRPO dataset using our base model. To elicit a wide range of reasoning pathways and outcomes for each problem, we set the sampling temperature to 1.0 for each run.

3. **Filtering for "Stochastic Correctness":** The correctness of every generated response is logged. After all rounds are complete, we filter this log to isolate the target samples. We select only those question-answer pairs that the model answered correctly in at least one round but incorrectly in at least one other round.

The rationale behind this selective filtering is to force the policy to learn to distinguish between successful and flawed reasoning pathways for the exact same problem. Training on these "boundary" cases ensures that the GRPO stage is dedicated to resolving ambiguity and reinforcing robust reasoning where it is most needed, leading to more significant and generalizable improvements in the model's core abilities.

### B.4 DETAILED FORMULATION OF THE ACCURACY REWARD

A core component of the GRPO framework is the **Continuous Accuracy Reward** ($R_{acc}$), which is designed to provide a dense, fine-grained signal for optimizing the model's numerical estimation capabilities. A simple binary reward (correct/incorrect) is often too sparse for reinforcement learning, as it fails to differentiate between a near-miss and a completely wrong answer. To overcome this, we designed a continuous reward function that recognizes and rewards "nearly correct" answers, thereby creating a smoother optimization landscape.

Our accuracy reward function provides a dense, informative, and well-behaved signal that is ideally suited for guiding our reinforcement learning process towards generating highly accurate and reliable numerical estimations.

### B.5 MORE TRAINING DETAILS

Our training process begins with the Qwen2.5-VL-7B-Instruct model as the foundation. Using the Swift framework, we perform SFT for 2 epochs on our 4.2k instruction-following dataset. In this stage we freeze the vision tower and the multimodal aligner while exclusively tuning the LLM's parameters. We set the learning rate to 1e-5 with a warm-up ratio of 0.05 and use an effective batch

size of 256. The SFT process is conducted on 8 NVIDIA H800 GPUs, utilizing bfloat16 precision and the DeepSpeed ZeRO-3 optimization strategy.

For the GRPO stage, we initialize the model with the checkpoint from the SFT phase and employ the GRPO algorithm on our 3.4k preference dataset. In this phase, we continue to freeze the vision tower but expand the scope of fine-tuning to include both the LLM and the multimodal aligner. The learning rate is reduced to 1e-6, again with a 0.05 warm-up ratio. For the rollout process, we use a generation batch size of 32 to create 4 completions per sample with a temperature of 1.0; the training itself uses an effective batch size of 64. The reward function is the composite of the Format and Continuous Accuracy rewards. The hardware and optimization setup remains consistent, utilizing 8 NVIDIA H800 GPUs with bfloat16 precision and DeepSpeed ZeRO-3.

## C  EVALUATION AND INFERENCE DETAILS

This section outlines the precise methodologies and inference settings used to evaluate all models and benchmarks, ensuring full reproducibility and fairness. Our protocols were designed by strictly adhering to the author-recommended settings and official evaluation scripts where available.

### C.1  GENERAL INFERENCE SETTINGS

All experiments reported in this paper were conducted using the default hyperparameters of each respective model, with no model-specific tuning performed at inference time. To ensure reproducibility, the random seed for all generation processes was fixed to 42, and the sampling temperature was set to 1.0. Our inference pipeline is built upon the vLLM framework, which provides efficient, high-throughput serving for Large Language Models.

### C.2  EVALUATION ON CHARTVRBENCH

Our evaluation on the proposed ChartVRBench employed different prompting strategies depending on the model type to ensure a fair and rigorous assessment.

**General MLLMs.**  To verify the visual reasoning capabilities of general-purpose models, we employed a structured Chain-of-Thought (CoT) prompt, shown in Figure 10. This prompt compels the model to first articulate its reasoning—by analyzing axes, data points, and context—before providing a final answer. The prompt enforces a strict separation between the step-by-step logic (output in '¡think¿' tags) and the concise final output (in an '¡answer¿' tag). This approach allows us to pinpoint the exact stage where a model's logic succeeds or fails, moving our analysis beyond simple accuracy metrics.

**Chart-Specific Models.**  In contrast, for models already fine-tuned on specific chart-related data formats (including ChartGemma, TinyChart, ChartInstruct, ChartVLM, and ChartLlama), we did not use our generalized CoT prompt. To elicit their best possible performance and establish the strongest baseline, we followed the official author-recommended procedures:

1. We cloned the official public repository for each model.

2. We utilized their provided, out-of-the-box inference scripts and default model weights without modification.

3. We fed the images and questions from our ChartVRBench test set directly into these scripts.

This methodology ensures that we are comparing our model against the most capable version of each specialized baseline.

### C.3  EVALUATION OF CHARTVR ON PUBLIC BENCHMARKS

To validate the generalization of *ChartVR*'s enhanced reasoning capabilities, we evaluated it against several standard public benchmarks, following the official protocol for each.

**CharXiv.** We utilized the official code and evaluation scripts from the CharXiv repository. We integrated our locally-deployed *ChartVR* as the model backend into their inference pipeline, keeping all other components (data loading, pre-processing, and scoring scripts) identical to the original setup.

**ChartBench.** Our evaluation followed a similar protocol using the complete pipeline from the official ChartBench repository. We generated predictions with our *ChartVR* model and fed the outputs directly into the official scoring script.

**ChartQAPro.** As the official repository provides a standalone evaluation script but not a full inference pipeline, we implemented a two-step process. First, we developed a script to generate predictions for the test set using a prompt that precisely replicated the template described in the ChartQAPro paper. Second, the resulting file of predictions was used as input for the official evaluation script to compute the final accuracy score.

## D INTERPRETATION OF HUMAN PERFORMANCE

It is important to note that human accuracy on this task is not 100%. This is primarily due to inherent tendencies in human visual estimation, for instance, individuals often gravitate towards estimating with round or integer values that appear close to the correct answer, rather than performing precise interpolation. Our analysis indicates that a 2% relative error tolerance is a reasonable threshold to account for these natural human inaccuracies.

Furthermore, performance varies significantly across chart types. For area charts, accuracy sees a substantial decline, because many questions require calculating the difference between the upper and lower boundaries of a shaded region, a task made considerably more difficult by the common absence of horizontal gridlines as visual aids. Similarly, for complex combo charts, lower performance can be attributed to cognitive factors, such as misinterpretation of the prompt or misunderstanding the intricate relationships between different chart components.

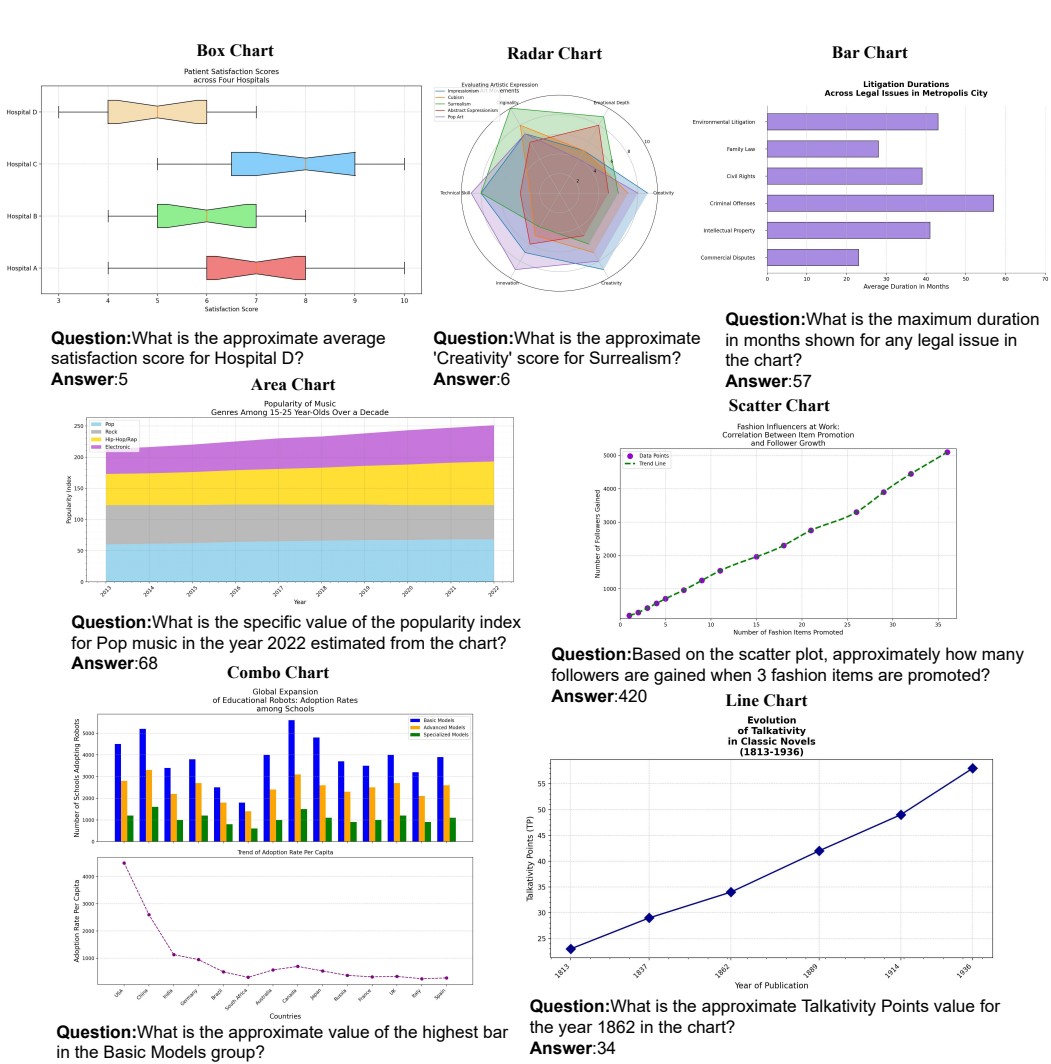

Figure 4: An overview of sample question-answer pairs for various synthetic chart types within the ChartVRBench dataset.

## Bar Chart

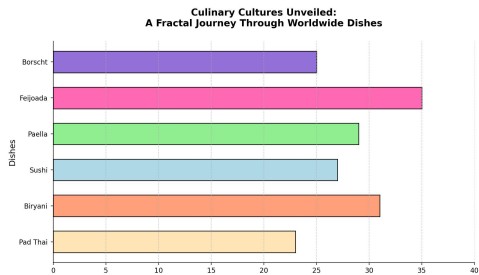

**Question:** What is the approximate sales volume for 'NetShop' in the chart? Please analyze based on the chart coordinates?
**Answer**: 85

**Question:** What is the approximate culinary complexity of Pad Thai as shown on the chart?
**Answer**: 23

## Line Chart

**Question:** What is the approximate brightness value of Star Alpha in August analyzed based on the line chart?
**Answer**: 85

**Question:** What is the specific step count value for the 20th day as shown in the graph?
**Answer**: 7191

## Combo Chart

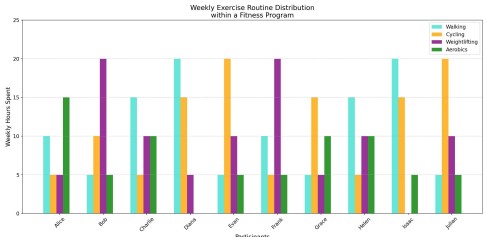

**Question:** What is the specific value of the weekly hours spent on Weightlifting by Charlie analyzed based on the coordinate axis?
**Answer**: 10

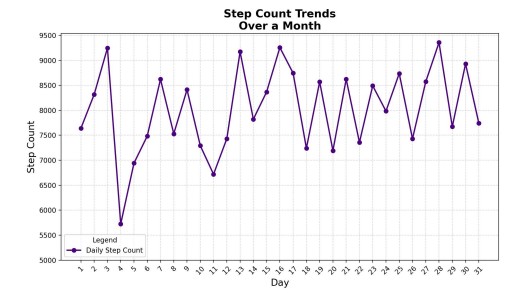

**Question:** What is the historical average rainfall in June based on the chart's coordinate axis?
**Answer**: 16

Figure 5: An overview of sample question-answer pairs for various complex synthetic chart types within the ChartVRBench dataset.

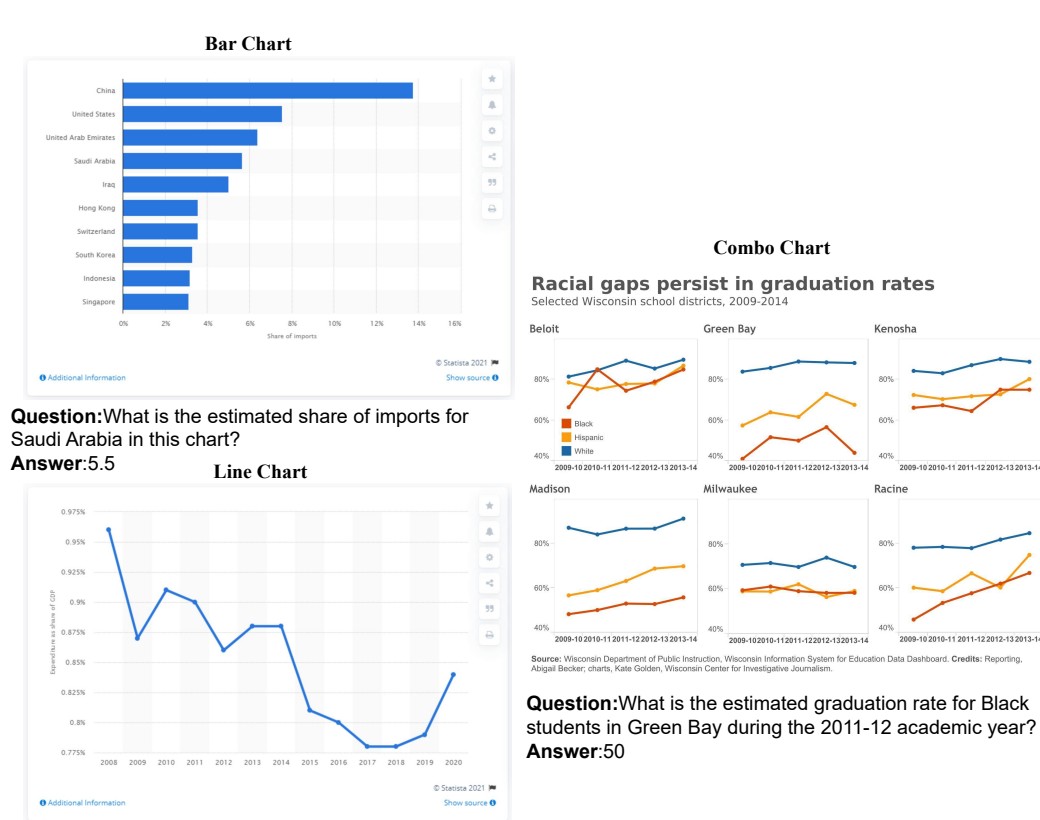

Figure 6: An overview of sample question-answer pairs for various real chart types within the ChartVRBench dataset.

**Distill CoT Data Prompt：**

=================================================

**ROLE**

=================================================

You are an expert vision-language analyst.
Your job is to look at the image, read the question, think step-by-step, and provide the final answer.

=================================================

**CRITICAL RULES (must follow all)**

=================================================

1. **Use ONLY the image and the question** when you think.
  └ The "Original Answer" is supplied **solely for self-checking**.
    └ NEVER quote, copy, hint at, or mention it in your reasoning.
      └ Forbidden words/phrases inside <think>:
    "original answer", "ground-truth", "GT", "correct answer", or the answer value itself.
2. Finish your full reasoning first, then decide your own answer
  (±2 % numerical tolerance is acceptable).
    └ If truly uncertain, output "uncertain" in <answer>.
3. Output exactly TWO tags in this order—nothing else:
<think>Your reasoning here</think>
<answer>Your final answer</answer>

=================================================

**INPUT FIELDS**

=================================================

Question    : {original_question}
Original Answer (for self-check only) : {original_answer}

=================================================

**EXAMPLE OUTPUT**

=================================================

<think>The point at "1900-present" lies slightly above the 90 tick; reading the y-axis, it corresponds to 93 on the popularity scale.</think>
<answer>93</answer>

Figure 7: The Prompt in Distilling the CoT Data

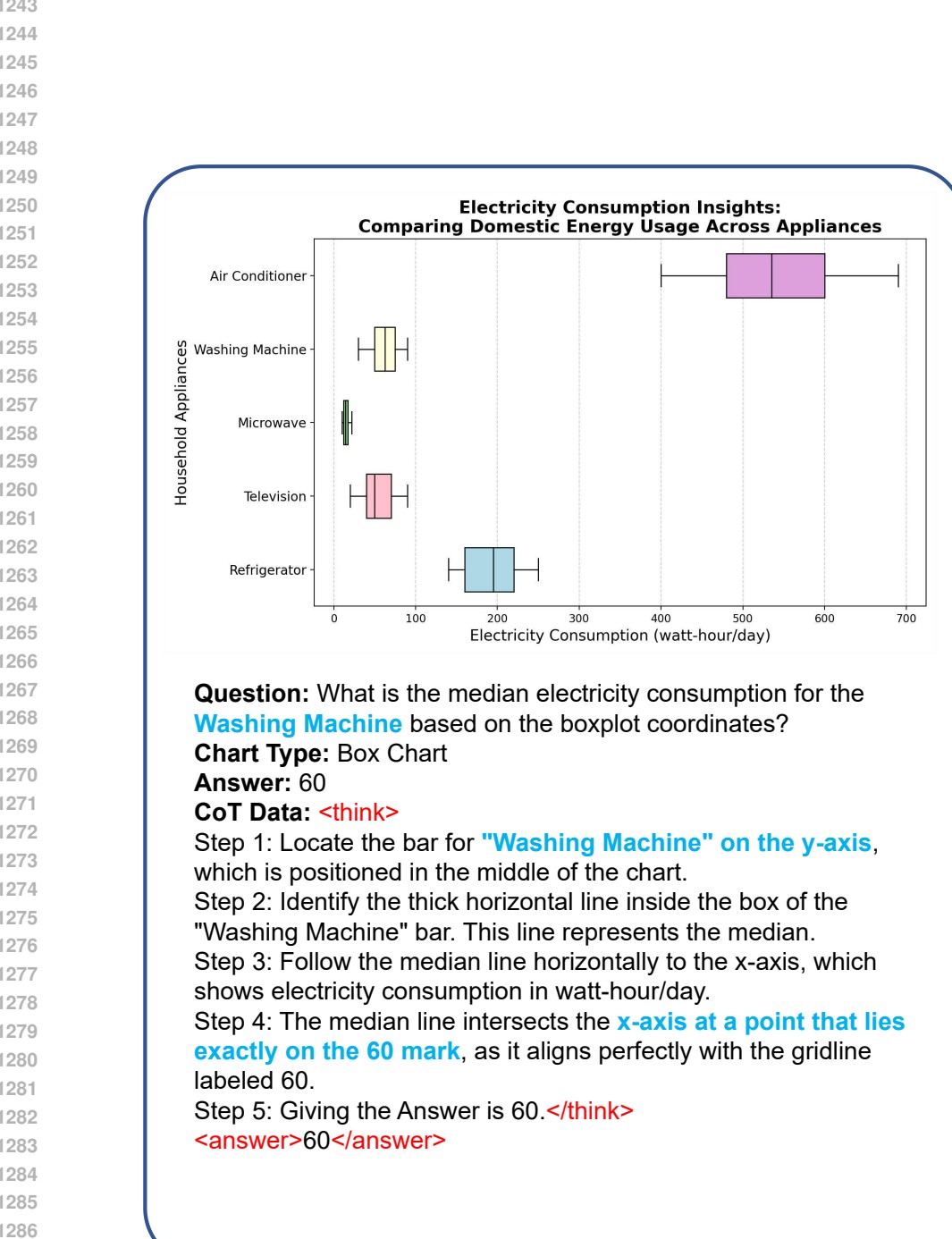

**Question:** What is the median electricity consumption for the **Washing Machine** based on the boxplot coordinates?
**Chart Type:** Box Chart
**Answer:** 60
**CoT Data:** <think>
Step 1: Locate the bar for **"Washing Machine" on the y-axis**, which is positioned in the middle of the chart.
Step 2: Identify the thick horizontal line inside the box of the "Washing Machine" bar. This line represents the median.
Step 3: Follow the median line horizontally to the x-axis, which shows electricity consumption in watt-hour/day.
Step 4: The median line intersects the **x-axis at a point that lies exactly on the 60 mark**, as it aligns perfectly with the gridline labeled 60.
Step 5: Giving the Answer is 60.</think>
<answer>60</answer>

Figure 8: Example 1 of CoT Dataset

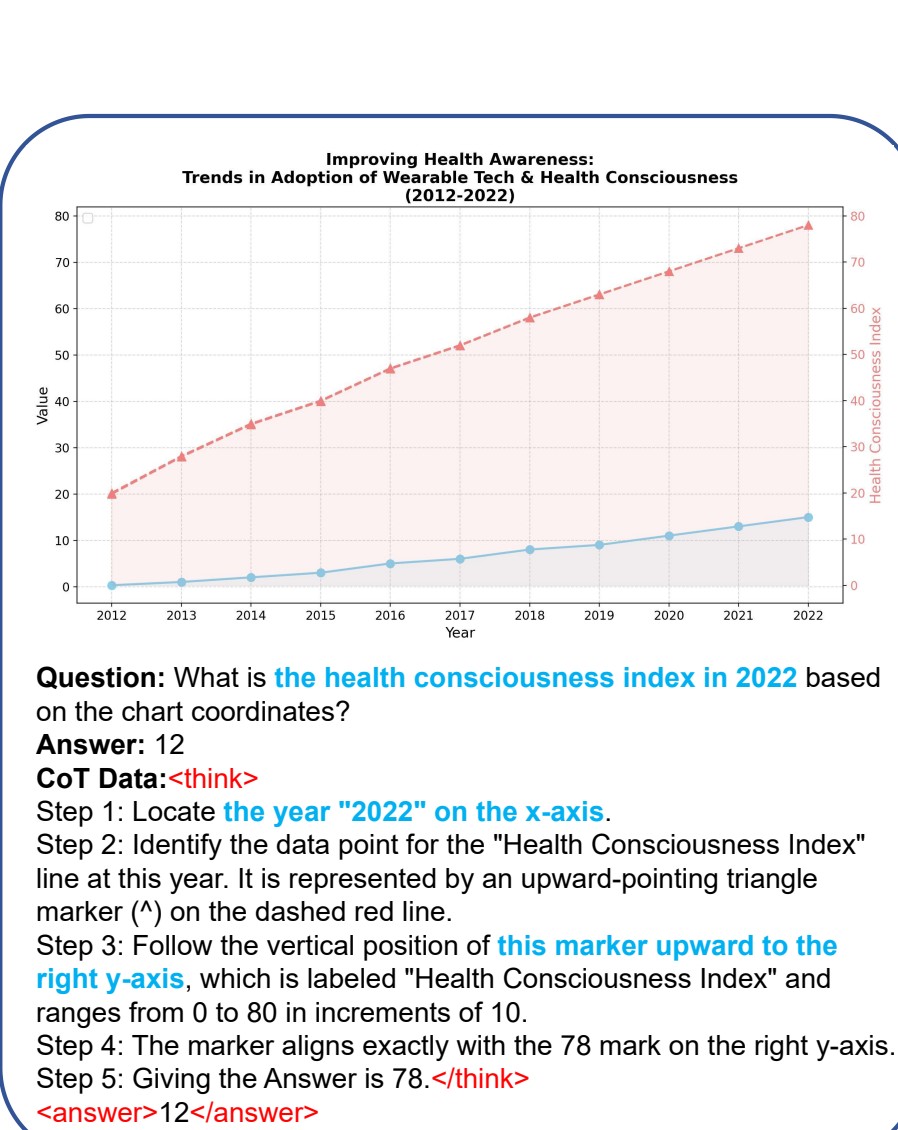

**Question:** What is **the health consciousness index in 2022** based on the chart coordinates?

**Answer:** 12

**CoT Data:**<think>

Step 1: Locate **the year "2022" on the x-axis**.

Step 2: Identify the data point for the "Health Consciousness Index" line at this year. It is represented by an upward-pointing triangle marker (^) on the dashed red line.

Step 3: Follow the vertical position of **this marker upward to the right y-axis**, which is labeled "Health Consciousness Index" and ranges from 0 to 80 in increments of 10.

Step 4: The marker aligns exactly with the 78 mark on the right y-axis.

Step 5: Giving the Answer is 78.</think>

<answer>12</answer>

Figure 9: Example 2 of CoT Dataset

# Chain of Thought(CoT) Prompt：

================================================

## ROLE

================================================

You are an expert vision-language analyst.
Your job is to look at the image, read the question, and provide a answer.

================================================

## CRITICAL RULES (must follow all)

================================================

1. **STEPBYSTEP THINKING:** You need to think step-by-step first before answering the question.Your thought process **(which you may output in the <think> tag)** should explicitly focus on:
 * **Axes:** What do the horizontal (X-axis) and vertical (Y-axis) represent? Note their labels, units, and scale.
 * **Data Points:** Locate the specific bars, points, lines, or other points relevant to the question.
 * **Context:** Read the chart's title, legend, and any other text to fully understand the context.
2. **FINAL ANSWER** Your output MUST contain the answer tag: `<answer>your answer</answer>`.
3. **STRICT FORMAT:** The answer inside the `<answer>` tag must be the final, concise result (e.g., a single number). Do not include explanations or units unless required by the chart's notation.

================================================

## INPUT FIELDS

================================================

Question     : {original_question}

Figure 10: CoT Prompting Reasoning Steps in ChartVRBench Evaluation

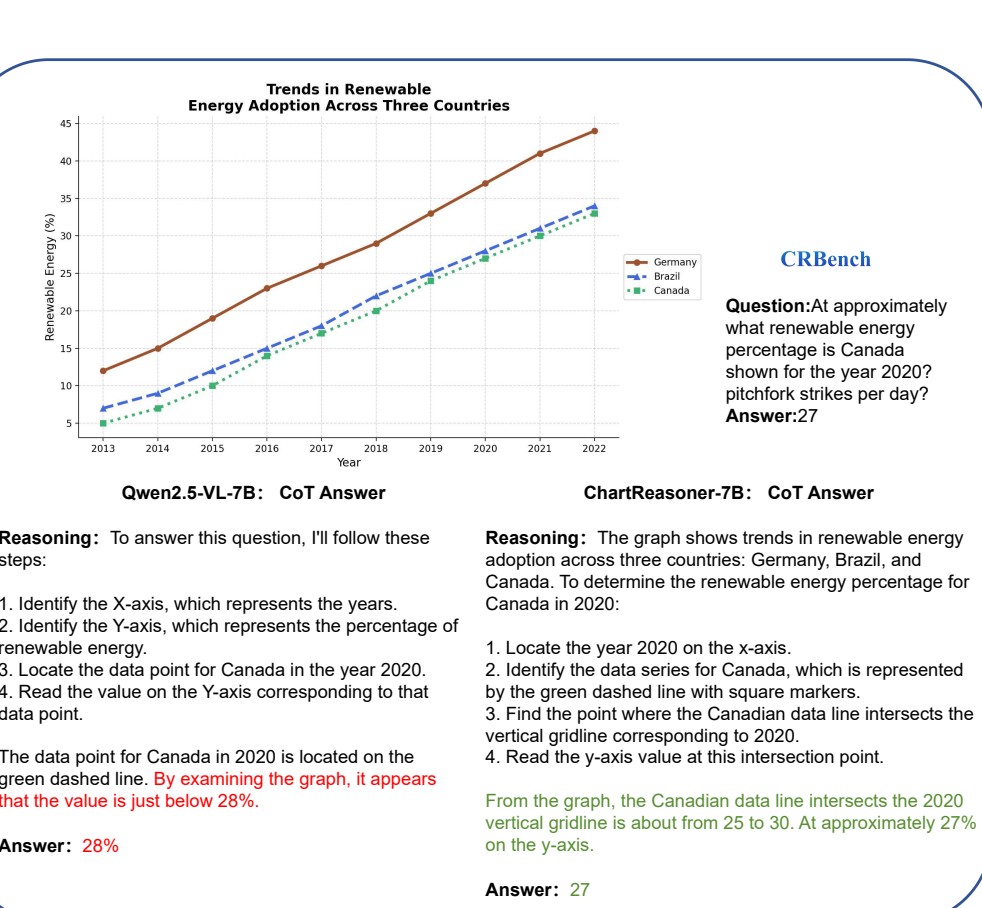

Figure 11: Example 1 from ChartVRBench comparing the CoT outputs of Qwen2.5-VL-7B and ChartVR-7B.

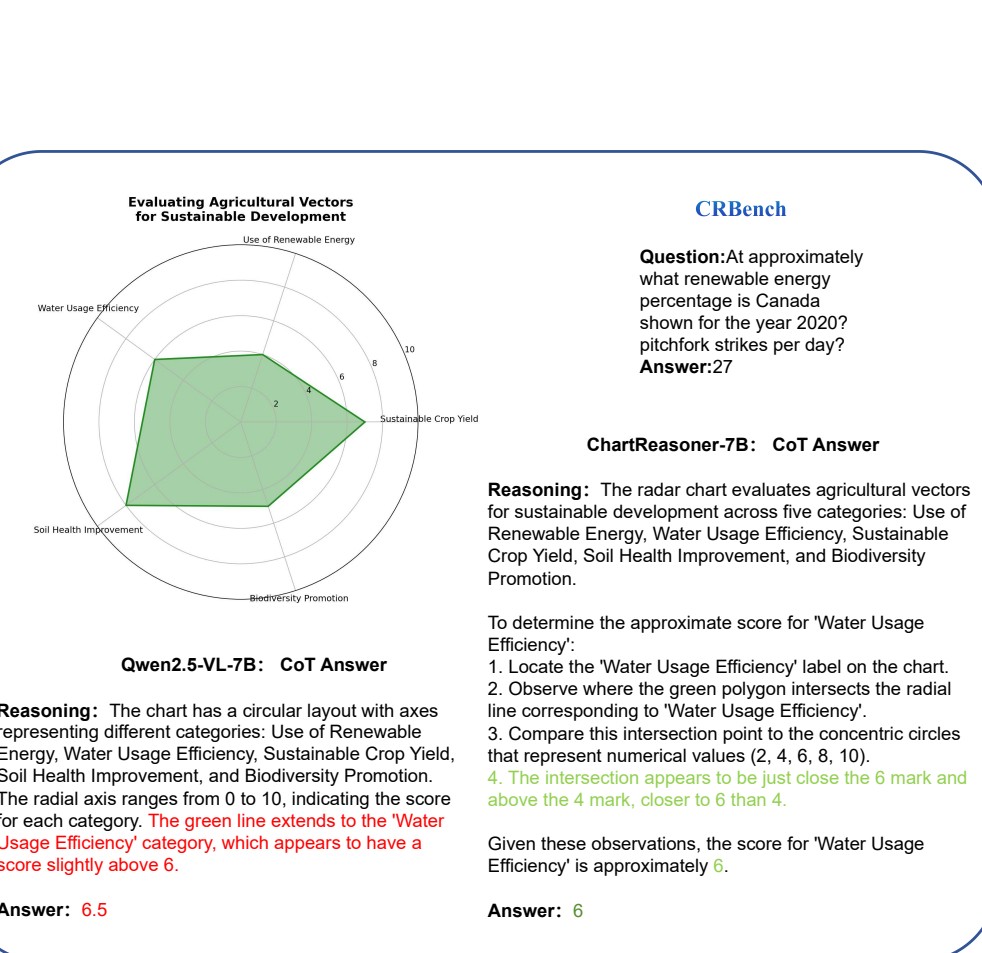

Figure 12: Example 2 from ChartVRBench comparing the CoT outputs of Qwen2.5-VL-7B and ChartVR-7B.

