# OpenReview forum: "Do MLLMs Really Understand the Charts?"
_ICLR.cc/2026/Conference — Submitted to ICLR 2026_

### Official Review · Reviewer_P44o · 2025-10-17

**Soundness:** 2
**Presentation:** 1
**Contribution:** 1
**Rating:** 2
**Confidence:** 5

**Summary:**

This paper investigates a critical failure mode in Multimodal Large Language Models (MLLMs): their inability to perform visual reasoning on charts that lack explicit numerical annotations. The authors argue that MLLMs excel at text recognition but fail at genuine visual reasoning, which they define as the ability to estimate numerical values from a chart's visual geometry (e.g., axes, scales, and positions).

To address this, the paper makes two primary contributions. First, it introduces ChartVRBench, a new benchmark specifically designed to isolate and evaluate this numerical estimation skill. The benchmark contains both synthetic and real-world charts, all deliberately stripped of explicit data labels. Second, it proposes ChartVR, a series of MLLMs trained with a novel two-stage Visual Reasoning Reinforcement Finetuning (VR-RFT) strategy. Stage 1 ("Activation") uses Supervised Fine-Tuning on distilled Chain-of-Thought (CoT) data to teach the model a structured reasoning process. Stage 2 ("Generalization") uses Group Relative Policy Optimization (GRPO), a reinforcement learning algorithm, to refine the model's precision, guided by a continuous quadratic accuracy reward function.

**Strengths:**

1. Excellent Problem Formulation: The paper's core premise is strong, clear, and important. It correctly identifies a key weakness in current MLLMs and frames a well-defined research question: are models reasoning or just recognizing? The focus on non-annotated charts is a methodologically sound way to isolate this specific capability.

2. Purpose-Built Benchmark: The creation of ChartVRBench, a benchmark designed specifically to test a single, well-defined skill (visual numerical estimation), is a good scientific practice. The inclusion of both synthetic and real-world data is a positive design choice.

**Weaknesses:**

1. High Risk of Data Contamination and Invalid Evaluation: The paper states that the training data for both the SFT and RL stages is derived directly from the ChartVRBench dataset itself. The SFT data consists of reasoning traces distilled from ChartVRBench's question-answer pairs, and the RL data consists of samples from ChartVRBench where the SFT model was inconsistent. The paper never explicitly states that it maintained a strict separation between the subset of ChartVRBench used for training data generation and the subset used for final evaluation. This strongly suggests that the model was trained on the same data distribution it was tested on, making the impressive results on ChartVRBench a measure of memorization or overfitting, not generalization. This potential train-test overlap invalidates the paper's primary empirical results.
2. Incremental Methodological Contribution: All the techniques are from prior works.
3. Narrow Evaluation Scope and Unconvincing Generalization Claims: The paper's claim that it imparts a "foundational and highly generalizable" reasoning ability is not sufficiently supported. The evaluation of generalization is restricted only to other chart-specific benchmarks (CharXiv, ChartBench, ChartQAPro). This demonstrates in-domain transfer, but it does not prove that the model has learned a general reasoning skill. To validate a claim about foundational reasoning, the evaluation should have included out-of-domain benchmarks that require similar geometric or logical skills, such as mathematical reasoning.
4. Concerns with Distilled Data Quality and Utility: The SFT stage relies on reasoning data distilled from a powerful teacher model (Qwen2.5-VL-32B). This introduces two issues: i) The student model's reasoning quality is capped by the teacher's ability. It is learning to mimic one specific, potentially flawed or suboptimal, reasoning style. ii) The very fact that a 32B model can generate these reasoning traces (even post-hoc) raises questions about the dataset's utility for training even more powerful models in the future. It creates a potential "ceiling effect" for the task itself.

**Questions:**

1. Can you please provide an unambiguous confirmation that a strict and held-out test set from ChartVRBench was used for evaluation, and that no part of this test set (images, questions, or answers) was used in any capacity during the SFT or RL data construction phases?

2. Given that VR-RFT is a pipeline of existing techniques (CoT-SFT, GRPO), could you clarify what you see as the core novel algorithmic contribution beyond the application to this specific task and the design of the reward function?

3. How can you substantiate the claim that the model has learned a "foundational" reasoning ability, rather than just a set of chart-specific interpretation skills? Why were no out-of-domain reasoning benchmarks (e.g., math, general VQA) included to test this claim of generalizability?

---

> ### Author Response · Authors · 2025-11-21
> **W1 & Q1: Clarification on Data Contamination and Train-Test Separation**
>
> We sincerely thank the reviewer for raising this critical point and identifying the confusion stemming from our manuscript’s phrasing. We apologize for the imprecise description in Appendix B.1, which inadvertently implied that the reasoning traces were distilled from the benchmark itself. We hereby provide an unambiguous confirmation that **ChartVRBench in its entirety is strictly a held-out test set used solely for evaluation**. The 43k SFT dataset and the 3.4k RL dataset were additionally constructed specifically for training purposes; while they share the same generation pipeline and domain scope as the benchmark to ensure consistency, they were generated as completely independent batches using distinct random seeds. Consequently, there is no overlap in images, questions, or answers between the training resources and the ChartVRBench evaluation set. We have conducted a rigorous verification to confirm zero data contamination.
>
> Furthermore, the fact that our model achieves impressive results on completely external benchmarks like CharXiv and ChartBench, as shown in Table 2, provides strong empirical evidence that the model has learned robust, generalizable visual reasoning skills rather than memorizing a chart-specific interpretation skill.
>
> We will update the manuscript to explicitly clarify the distinction between the training corpora and the evaluation benchmark.

---

> ### Author Response · Authors · 2025-11-21
> **W2 & Q2: Methodological Novelty and Core Contribution**
>
> We thank the reviewer for this thoughtful critique regarding the methodological novelty of our work. We argue that our contribution extends beyond the mere application of existing components; it lies in the rigorous refinement of data synthesis techniques and the strategic formulation of Visual Reasoning Reinforcement Finetuning (VR-RFT) to target a foundational cognitive gap in MLLMs.
>
> First, regarding the data generation pipeline, while we utilize established foundations like Code-as-Intermediary Translation and Self-Instruct, our contribution lies in the significant refinement and diversification of these techniques to ensure realism. We went beyond basic generation by implementing strict prompt engineering constraints, such as prohibiting integer alignment with axes and enforcing non-monotonic trends, to force the model to perform genuine visual interpolation rather than simple pattern matching. As detailed in Appendix A, we significantly expanded the scope to cover 38 distinct topics and multiple chart types to capture the long-tail distribution of real-world data. Furthermore, unlike benchmarks that rely solely on synthetic data, ChartVRBench incorporates a significant portion of real-world charts manually curated from diverse online sources. This hybrid approach ensures that our evaluation metrics reflect performance on actual, in-the-wild data distributions rather than just synthetic artifacts.
>
> Second, regarding the algorithmic contribution of VR-RFT, we view the precise definition and resolution of the numerical value estimation task as a core contribution. We posit that visual estimation is a foundational visual reasoning primitive—an atomic skill required for higher-level chart tasks. Our approach goes beyond standard RL applications by implementing a novel data curation strategy for the RL stage that targets samples of "stochastic correctness"—specifically, problems where the SFT-tuned model exhibits inconsistent performance (solving them correctly only in some attempts). By concentrating training on these borderline cases with higher-resolution images, we maximize learning efficiency and force the model to refine its visual interpretation skills where it is most uncertain. Coupled with our novel Continuous Quadratic Accuracy Reward, this targeted intervention effectively cultivates the foundational capability of precise estimation. The validity of this hypothesis is borne out by our experimental results: by solving this specific estimation task, our model achieves impressive generalization across broad, multi-task benchmarks like CharXiv and ChartQAPro. This demonstrates that VR-RFT does not just teach a narrow task, but instills a robust reasoning capability that transfers to general chart understanding, a finding that offers significant value to the community beyond the algorithm itself.

---

> ### Author Response · Authors · 2025-11-21
> **W3 & Q3: Scope of "Foundational" Reasoning and Generalization Claims**
>
> We appreciate the reviewer for identifying this potential ambiguity in our terminology. We wish to clarify that our use of the terms "foundational" and "generalizable" was intended to be understood strictly within the specific domain of **chart understanding**, rather than as a claim about imparting a universal reasoning ability applicable to general mathematics or open-ended VQA.
>
> Our central thesis is that **visual numerical estimation**—the ability to correctly map visual geometry (e.g., position, length, angle) to numerical values—is the core implementation of **visual reasoning** in the context of charts. By cultivating this specific capability through our VR-RFT strategy, we aim to equip the model with a robust, generalized ability to interpret chart data fidelity, regardless of the chart style or data distribution. Consequently, the "generalization" we claim is the successful transfer of this cultivated visual reasoning skill across the diverse landscape of chart understanding tasks. This is evidenced by the fact that our model, trained on atomic estimation tasks, achieves significant performance gains on broad, complex benchmarks like CharXiv and ChartQAPro. These benchmarks contain a wide variety of chart types and visual distributions that differ significantly from our training data. The model's success on these external datasets demonstrates that it has learned a generalized visual interpretation capability that is effective across the entire chart domain, rather than simply overfitting to specific estimation problems.

---

> ### Author Response · Authors · 2025-11-21
> **W4 : Teacher Quality and "Ceiling Effect" in Distillation**
>
> We thank the reviewer for raising this important question regarding the limitations of distillation and the quality of the teacher model. We address these concerns by discussing our specific choice of teacher and referencing recent literature that challenges the assumption of a hard ceiling in reasoning distillation.
>
> We deliberately selected Qwen2.5-VL-32B-Instruct not merely for its scale, but for its specific post-training optimizations. As detailed in its model card, this model underwent rigorous reinforcement learning to enhance clarity of formatting and logical reasoning for objective queries. This makes it an exceptionally high-fidelity source for generating structured Chain-of-Thought traces. By filtering for high-confidence, logically sound traces during our data curation as detailed in Appendix B.1, we mitigate the risk of the student learning from suboptimal or hallucinated reasoning styles.
>
> Additionally, contrary to the traditional view that a student model is strictly capped by its teacher, recent research have already demonstrated that this ceiling effect often disappears when distilling reasoning processes rather than just final answers. For instance, [1] demonstrated that smaller student models can outperform massive teachers when trained on rationales rather than just labels. This occurs because the student learns the algorithm of the problem-solving process, allowing it to generalize better than the teacher, which may rely on memorized heuristics. Furthermore, [2] have shown that weak supervisors can elicit strong capabilities in student models, effectively proving that the teacher's performance does not set a hard upper bound. In fact, distillation often acts as a denoising process; if a teacher outputs correct reasoning most of the time but occasionally hallucinates, a student trained on a curated, verified subset can learn the robust signal while discarding the teacher's stochastic noise, effectively surpassing the teacher's average reliability.
>
> Finally, we emphasize that our primary contribution is not just the static dataset, but the generalizable methodology. Our pipeline is teacher-agnostic. As more powerful reasoning models emerge, they can be plugged into our framework to generate even higher-quality CoT data, continuously raising the ceiling. Thus, the utility of our method scales with the progress of the broader field.
>
> ---
>
> [1] Hsieh, Cheng-Yu, et al. “Distilling step-by-step! outperforming larger language models with less training data and smaller model sizes.” In Proceedings of the 61st Annual Meeting of the Association for Computational Linguistics (Volume 1: Long Papers), pp. 10003–10017, 2023.
>
> [2] Burns, Collin, et al. “Weak-to-Strong Generalization: Eliciting Strong Capabilities With Weak Supervision.” In The Twelfth International Conference on Learning Representations (ICLR), 2024.

---

### Official Review · Reviewer_M2Y6 · 2025-10-27

**Soundness:** 2
**Presentation:** 2
**Contribution:** 2
**Rating:** 4
**Confidence:** 4

**Summary:**

The paper examines whether Multimodal Large Language Models (MLLMs) truly comprehend charts, especially non-annotated ones requiring visual estimation of numerical values. It introduces ChartVRBench, a benchmark with 2,453 Q&A pairs across seven chart types, using synthetic and real-world data to test visual reasoning. The authors propose ChartVR-3B/7B, trained with a two-stage Visual Reasoning Reinforcement Finetuning (VR-RFT) method—Chain-of-Thought Supervised Finetuning (CoT-SFT) followed by Group Relative Policy Optimization (GRPO) with a continuous accuracy reward. Experiments show ChartVR outperforming models like Gemini-2.5-Flash on ChartVRBench and generalizing well to other benchmarks.

**Strengths:**

**Important Topic**: Chart understanding is a crucial area for advancing MLLM applications in real-world data analysis, such as scientific papers and financial reports.

**Comprehensive Training Approach**: The VR-RFT strategy effectively combines SFT and RL, with the continuous quadratic accuracy reward delivering a dense, fine-grained signal to refine visual estimation precision.

**Strong Empirical Results and Generalization**: ChartVR demonstrates superior performance on the new benchmark and transfers well to diverse public datasets, suggesting the cultivated visual reasoning is foundational and not task-specific. The paper includes qualitative case studies highlighting improved reasoning chains.

**Weaknesses:**

**Limited Novelty**:  I think this is the main weakness of this paper. The core idea of visual reasoning on non-annotated charts is not entirely new; benchmarks like PlotQA (cited in the paper, but several years before) already include such data without explicit annotations. Also, the synthetic data generation relies on established methods like Code-as-Intermediary Translation and Self-Instruct, reducing the perceived innovation in dataset curation.

**Insufficient Comparison to Recent SOTA Models**: The evaluation lacks direct comparisons with stronger open-source chart-specific models like Chart-R1, which employs similar CoT supervision and RL strategies to achieve state-of-the-art results on chart understanding benchmarks—potentially undermining claims of superiority.

**Questions:**

See Cons.

---

> ### Author Response · Authors · 2025-11-21
> **W1: Novelty of ChartVRBench and Data Curation**
>
> While we acknowledge that the broad concept of non-annotated charts exists in prior literature, we argue that ChartVRBench offers significant novelty in its methodological precision and visual complexity, addressing critical limitations in older benchmarks like PlotQA.
>
> First, regarding the comparison with PlotQA, we emphasize that this benchmark was introduced in 2019, before the era of modern MLLMs. Consequently, its charts are often visually simplified and homogeneous, failing to represent the growing visual complexity of data visualizations used in recent years. More importantly, PlotQA predominantly focuses on complex QA tasks that conflate visual perception with textual reasoning. For example, answering a typical PlotQA question like "What is the average number of Hispanic students in school?" requires a dependent chain of visual estimation followed by arithmetic calculation. If a modern MLLM fails this task, it is impossible to determine whether the error stemmed from misreading the chart's geometry or a failure in arithmetic reasoning. ChartVRBench is novel because it effectively decouples these skills, focusing exclusively on Visual Reasoning—drawing inferences strictly from a chart's geometric properties. This provides a targeted diagnostic signal for visual hallucinations that older benchmarks cannot offer.
>
> And to clearly demonstrate how our benchmark compares to contemporary works, we present a comparison here:
>
> |  | **EvoChart** | **ChartBench** | **CharXiv** | **ChartQAPro** | **ChartMuseum** | **ChartVRBench (Ours)** |
> | --- | --- | --- | --- | --- | --- | --- |
> | **Real-World Charts** | $\checkmark$ | $\times$ | $\checkmark$ | $\checkmark$ | $\checkmark$ | $\checkmark$ |
> | **Broad Topic Coverage** | $\times$ | $\times$ | $\triangle$ | $\triangle$ | $\triangle$ | $\checkmark$ |
> | **Focus on Non-Annotated Charts** | $\times$ | $\checkmark$ | $\times$ | $\times$ | $\times$ | $\checkmark$ |
> | **Isolates Visual Reasoning** | $\times$ | $\triangle$ | $\times$ | $\times$ | $\checkmark$ | $\checkmark$ |
>
> *Symbols: $\checkmark$ Fully Supported / High Quality; $\triangle$ Partially Supported / Mixed; $\times$ Not Supported / Low Quality.*
>
> Second, regarding the data generation pipeline, while we utilize established foundations like Code-as-Intermediary Translation and Self-Instruct, our contribution lies in the significant refinement and diversification of these techniques to ensure realism. We went beyond basic generation by implementing strict prompt engineering constraints, such as prohibiting integer alignment with axes and enforcing non-monotonic trends, to force the model to perform genuine visual interpolation rather than simple pattern matching. As detailed in Appendix A, we significantly expanded the scope to cover 38 distinct topics and multiple chart types to capture the long-tail distribution of real-world data. Furthermore, unlike benchmarks that rely solely on synthetic data, ChartVRBench incorporates a significant portion of real-world charts manually curated from diverse online sources. This hybrid approach ensures that our evaluation metrics reflect performance on actual, in-the-wild data distributions rather than just synthetic artifacts.

---

> ### Author Response · Authors · 2025-11-21
> **W2: Comparison with Concurrent Work (Chart-R1)**
>
> We appreciate the reviewer bringing Chart-R1 to our attention and recognize it as a highly relevant work that shares the community's growing interest in applying Reinforcement Learning to chart understanding.
>
> However, we respectfully point out that Chart-R1 is a *contemporaneous work* that is currently under review for ICLR 2026 alongside our submission. According to the ICLR 2026 Reviewer Guide:
>
> > "Note that arXiv is not considered a peer-reviewed venue. As such, authors are not required to compare to papers solely on arXiv: they may be excused for not knowing about papers not published in peer-reviewed conference proceedings or journals."
> >
>
> Since Chart-R1 has not yet been published in a peer-reviewed venue and exists solely as a preprint/concurrent submission, strictly speaking, a direct empirical comparison is not required by conference policy.
>
> To address the reviewer's underlying concern about evaluating against stronger open-source chart-specific models using similar strategies, we explicitly included Bespoke-MiniChart-7B in our evaluation as shown in Table 2. This serves as a rigorous head-to-head control because both Bespoke-MiniChart-7B and our ChartVR-7B are built on the exact same Qwen2.5-VL-7B backbone. Despite sharing the same powerful foundation, ChartVR-7B consistently outperforms Bespoke-MiniChart-7B. Specifically, we achieve 72.20% on ChartVRBench compared to 68.16% and demonstrate superior generalization on public benchmarks like CharXiv, ChartBench, and ChartQAPro. This performance gap effectively showcases the superiority of our VR-RFT training strategy.

---

> > ### Author Response · Authors · 2025-12-03
> > **Follow-up Experiment on Chart-R1**
> >
> > To address the reviewer's interest and demonstrate the robustness of our method, we have conducted additional experiments to compare ChartVR directly against Chart-R1.
> >
> > We reproduced the results for Chart-R1 (7B) and compared them with ChartVR-7B across four benchmarks: our proposed ChartVRBench, plus CharXiv, ChartBench, and ChartQAPro. The results (also added to the manuscript in Table 2) are summarized below:
> >
> > | **Model** | **ChartVRBench** | **ChartBench** | **ChartQAPro** | **CharXiv** |
> > | --- | --- | --- | --- | --- |
> > | **Chart-R1 (7B)** | 65.80 | 15.71 | 31.81 | **50.00** |
> > | **ChartVR-7B (Ours)** | **72.20** | **45.34** | **41.79** | 43.40 |
> >
> > ChartVR-7B outperforms Chart-R1 on ChartVRBench (+6.4%), validating the effectiveness of our VR-RFT strategy for atomic visual reasoning. Furthermore, ChartVR demonstrates superior generalization, surpassing Chart-R1 by nearly 30% on ChartBench and ~10% on ChartQAPro. While Chart-R1 appears over-optimized for scientific charts (CharXiv), ChartVR maintains robust, state-of-the-art performance across diverse domains.

---

### Official Review · Reviewer_kvg2 · 2025-10-30

**Soundness:** 3
**Presentation:** 3
**Contribution:** 3
**Rating:** 6
**Confidence:** 3

**Summary:**

This paper studies the problem that MLLMs fails to undertand charts without annotations. A specific benchmark ChartVRBench is developed to comprehensively evaluate the ablites of MLLMs in such a scenario. Besides, to boost the performance of MLLMs in undertanding annotation-free charts, ChartVR is proposed.

**Strengths:**

1. The paper is well-organized, clearly introducing the background and motivation.
2. The motivation and problem studied in this paper are significant, highlighting current MLLMs could not understand charts without annoations accurately.
3. The benchmark is necessary for the field of chart understanding, which measures the ability of MLLMs in chart understanding in a specific scenario.
4. The performance of ChartVR on both ChartVRBench and public general chart understanding benchmark is great.

**Weaknesses:**

1. In Table 2, since MLLMs with size of 3B are compared, the performance of ChartVR-3B on public chart understanding benchmarks is supposed to be compared.
2. The performance of MLLMs on Radar charts is poor and far from human baseline, does it mean the reasoning pattern could not generalize to this type?

**Questions:**

See the weaknesses.

---

> ### Author Response · Authors · 2025-11-21
> **W2: Weak Generalization on Radar Charts**
>
> We acknowledge the reviewer's insightful observation regarding the performance gap on Radar charts. We agree that the reasoning pattern cultivated by our method shows limited generalization to this specific chart type, and we attribute this to the fundamental geometric disparity between coordinate systems.
>
> Standard charts, such as bar, line, and scatter plots, rely on a Cartesian coordinate system, where value estimation involves projecting a point orthogonally to an axis. In contrast, Radar charts utilize a Polar coordinate system, which requires the model to estimate a radius relative to a center point across angular axes. Our analysis suggests that current MLLMs, including powerful proprietary models, exhibit a strong inductive bias toward Cartesian layouts due to their dominance in pre-training data. Consequently, while our VR-RFT strategy successfully enhanced visual reasoning across diverse Cartesian types (transferring effectively from simple Bar charts to complex Combo and Scatter plots), the transfer to the radically different spatial structure of Polar coordinates remains a significant challenge. This represents a common limitation in the field rather than a specific failing of our training method, and we view bridging this Cartesian-to-Polar gap as a critical direction for future research.

---

> ### Author Response · Authors · 2025-11-21
> **W1: Performance of ChartVR-3B on Public Benchmarks**
>
> We have conducted the additional experiments and updated Table 2 in our manuscript to include the performance of ChartVR-3B. The results demonstrate that ChartVR-3B achieves superior performance compared to other chart-specific models of similar size, such as ChartGemma-3B and TinyChart-3B, across all these public benchmarks.

---

### Official Review · Reviewer_zvwv · 2025-10-31

**Soundness:** 2
**Presentation:** 3
**Contribution:** 2
**Rating:** 2
**Confidence:** 5

**Summary:**

This paper introduces ChartVRBench, a benchmark designed to isolate and evaluate visual reasoning capability in chart understanding. The authors proposed ChartVR, a series of LVLM with significantly enhanced visual reasoning capabilities for chart understanding. The paper tackles an important problem at the intersection of LVLM and data visualization.

**Strengths:**

- The paper is well-written and easy to follow, with a clear structure.
- The motivation behind the work is interesting and well-presented.
- The proposed method (ChartVR) is technically sound and clearly described.

**Weaknesses:**

1. The paper is missing a discussion and analysis of the proposed benchmark. This paper claims the benchmark, ChartVRBench, is part of its contribution. However, there is limited analysis of the benchmark's quality or comparison with existing benchmarks.
2. The chart data in the benchmark are mostly synthesized by an LVLM. Without assessing its quality, the performance on the benchmark cannot be supported. Evaluation, such as using human evaluators to check the correctness of the generated answers and the diversity of the generated questions, is highly recommended. Also, using an LVLM to generate data can introduce its own biases; for example, generated charts tend to have monotonic trends (only going upward or downward). One can observe in Figure 5 that the charts seem to have only simple trends, which is largely different from the real-world distribution. How is this addressed in the data generation pipeline? Is there any post-filtering or evaluation to ensure correctness and diversity?
3. The benchmark features charts without annotations. However, existing benchmarks (such as ChartXiv, ChartBench, and EvoChart) also contain charts without annotations. What is the difference between the proposed benchmark and these existing works? What are the novel benefits that this benchmark brings to the community? Is the data higher quality? Does the data cover more chart types? Or is the data aligned more closely with the real-world distribution?
4. A head-to-head comparison is needed to show the effectiveness of the proposed solution. The training of ChartVR involves an RL algorithm (GRPO), which is different from existing supervised finetuning techniques. A head-to-head comparison with previous methods on published benchmarks is necessary to support the effectiveness of the proposed method. For example, the comparison in the table is unfair: ChartVR-7B is built on Qwen 2.5VL, which is a much stronger backbone model than those used in existing approaches. A fairer comparison would involve keeping the backbone models the same as in previous work to showcase the effectiveness of both the proposed dataset and the training algorithm. Additionally, you should fix the model and data for both previous works and your model, then train using the previous SFT algorithm and your RL training to showcase the effectiveness of the proposed training strategies. Without these head-to-head comparisons, I cannot be convinced of the approach's effectiveness.

**Questions:**

- How are the intermediate steps of the reasoning evaluated on ChartVRBench? I cannot find this in Section 3.2 of the evaluation protocol.

---

> ### Author Response · Authors · 2025-11-21
> **W1: Lack of Benchmark Discussion and Comparison:**
>
> We thank the reviewer for highlighting the need for a more detailed analysis and comparison. While existing benchmarks have significantly advanced the field, they predominantly focus on general high-level Question Answering (QA) where visual reasoning is often conflated with textual extraction (OCR) and logical reasoning. ChartVRBench fills a critical gap by being the first to strictly isolate the visual reasoning capability of numerical value estimation on non-annotated charts, preventing models from relying on text recognition "shortcuts."
>
> To clearly demonstrate how our benchmark compares to contemporary works, we present a comparison here:
>
> |  | **EvoChart[1]** | **ChartBench [2]** | **CharXiv [3]** | **ChartQAPro [4]** | **ChartMuseum [5]** | **ChartVRBench (Ours)** |
> | --- | --- | --- | --- | --- | --- | --- |
> | **Real-World Charts** | $\checkmark$ | $\times$ | $\checkmark$ | $\checkmark$ | $\checkmark$ | $\checkmark$ |
> | **Broad Topic Coverage** | $\times$ | $\times$ | $\triangle$ | $\triangle$ | $\triangle$ | $\checkmark$ |
> | **Focus on Non-Annotated Charts** | $\times$ | $\checkmark$ | $\times$ | $\times$ | $\times$ | $\checkmark$ |
> | **Isolates Visual Reasoning** | $\times$ | $\triangle$ | $\times$ | $\times$ | $\checkmark$ | $\checkmark$ |
>
> *Symbols: $\checkmark$ Fully Supported / High Quality; $\triangle$ Partially Supported / Mixed; $\times$ Not Supported / Low Quality.*
>
> **Detailed Differences:**
>
> - **Vs. EvoChart:** While EvoChart contains a very small number of real-world non-annotated charts, it is not the focus of this benchmark. Moreover, from the example of EvoChart, the visual reasoning is logical reasoning extracted through OCR capabilities, rather than true visual reasoning.
> - **Vs. ChartBench:** While ChartBench addresses the problem of non-annotated charts, it relies on purely synthetic data with limited visual diversity (e.g., homogenous matplotlib styles). As discussed in Section 2.1, this lack of visual complexity fails to test the robustness required for real-world applications. ChartVRBench employs a hybrid approach, combining diverse synthetic layouts with real-world data to ensure both control and realism.
> - **Vs. CharXiv:** While CharXiv provides high-quality real-world charts, it is strictly limited to the scientific domain, i.e. solely from arXiv papers. ChartVRBench is composed of synthetic charts covering 38 distinct topics (including finance, healthcare, agriculture, etc.) and real-world charts from multiple platforms, ensuring significantly broader applicability.
> - **Vs. ChartQAPro:** ChartQAPro focuses on high-level reasoning (e.g., hypothetical, conversational QA). This focus often conflates visual reasoning with textual/logical reasoning. If a model fails, it is difficult to diagnose whether the error stems from misinterpreting the geometry of charts or failing the logical deduction. ChartVRBench isolates the atomic visual step.
> - **Vs. ChartMuseum:** While ChartMuseum distinguishes between visual and textual reasoning, it has a critical limitation: it deliberately omits chart types where values are not precisely aligned with axis ticks. In the real world, data points rarely align perfectly with grid lines, requiring numerical value estimation. ChartVRBench specifically targets this challenging "unaligned" scenario, whereas ChartMuseum restricts itself to the simpler "extraction" of aligned values.
>
> ---
> [1] Huang, Muyue, et al “EvoChart: A Benchmark and a Self-Training Approach Towards Real-World Chart Understanding.” Proceedings of the AAAI Conference on Artificial Intelligence, 39(4): 3680–3688, 2025.
>
> [2] Xu, Zhengzhuo, et al. *“ChartBench: A Benchmark for Complex Visual Reasoning in Charts.”* arXiv preprint arXiv:2312.15915 (2024).
>
> [3] Wang, Zirui, et al. *“CharXiv: Charting Gaps in Realistic Chart Understanding in Multimodal LLMs.”* In *The Thirty-eighth Conference on Neural Information Processing Systems Datasets and Benchmarks Track* (NeurIPS D&B), 2024.
>
> [4] Masry, Ahmed, et al. *“ChartQAPro: A More Diverse and Challenging Benchmark for Chart Question Answering.”* In *Findings of the Association for Computational Linguistics: ACL 2025*, pp. 19123–19151. ACL, Vienna, 2025.
>
> [5] Tang, Liyan, et al. *“ChartMuseum: Testing Visual Reasoning Capabilities of Large Vision-Language Models.”* In *The Thirty-ninth Annual Conference on Neural Information Processing Systems Datasets and Benchmarks Track* (NeurIPS D&B), 2025.

---

> ### Author Response · Authors · 2025-11-21
> **W2: Benchmark Quality and Bias**
>
> We thank the reviewer for this scrutiny and fully agree that synthetic data can suffer from hallucinations or simplistic patterns if left unchecked. We addressed these risks through a strict human-in-the-loop verification process and constraint-based prompting mechanisms.
>
> Regarding the correctness of the data, as stated in Section 3.1 and Appendix A.3.2, the entire synthesized benchmark underwent a rigorous manual review process where human annotators verified **100% of the samples**. This served as a strict post-filtering step where annotators were instructed to validate visual fidelity and discard any chart with rendering artifacts or factually incorrect Q&A pairs. Consequently, the performance metrics reported on ChartVRBench reflect genuine model capabilities.
>
> To address the concern regarding generative bias and monotonic trends, our data generation pipeline was explicitly engineered to inject entropy and complexity into the charts. We utilized specific structural constraints in our prompts to prevent the generation of simplistic linear trends or values that align too perfectly with grid lines.
>
> For instance, our generation prompts explicitly state that:
>
> > You must not use random() to construct the data... when constructing data, try to use integers, but don't make all values multiples of five and ten, as this can prevent all values from aligning with the coordinate axis.
> >
>
> Furthermore, we explicitly prompted the model to generate varied data patterns, including fluctuations, multi-modal distributions, and outliers; the examples in Figure 5 were selected primarily for visual clarity in the paper and do not represent the full statistical complexity of the benchmark. This structural diversity is further reinforced by our coverage of 38 distinct topics, ensuring the data reflects the varying magnitudes and distributions found in diverse real-world domains.
>
> We will include a clearer breakdown of these constraints and additional examples of complex, non-monotonic charts in the Appendix to better illustrate the true distribution.

---

> ### Author Response · Authors · 2025-11-21
> **W3: The Novelty of ChartVRBench**
>
> We have partially addressed the comparison with existing benchmarks in our response to W1, particularly regarding data sources, topic diversity, and the feature-wise breakdown. Here, we wish to elaborate on the fundamental methodological distinction that sets ChartVRBench apart: the strict isolation of Visual Reasoning from Textual Reasoning.
>
> While prior benchmarks (such as ChartXiv, ChartBench, and EvoChart) include non-annotated charts, they often conflate these two distinct capabilities. We define them as follows:
>
> - **Visual Reasoning** involves drawing inferences strictly from a chart's visual properties, which arise more naturally from visual comparison than from reasoning over extracted text. For instance, identifying the value in a non-annotated chart is a quintessential visual task, as it relies on perceiving the overall geometric pattern (e.g., the height of a bar relative to an axis) rather than extracting and calculating numeric values via text processing.
> - **Textual Reasoning** involves logical, arithmetic, or comparative operations on information that has *already* been textually extracted from the chart. For instance, after using OCR to extract the specific numerical values of several bars, performing arithmetic operations to calculate their total sum is a task of textual reasoning.
>
> With these definitions in mind, the primary novelty of our work is its specific focus on **isolating and evaluating pure visual reasoning**. Existing benchmarks often present questions that require a complex chain of both skills, making it difficult to pinpoint the source of model failures. For example, consider the task: *"Calculate the sum of all bars."* A model's success requires both correct visual reasoning (identifying bars and estimating their values from the axis) and accurate textual/arithmetic reasoning (summing the numbers). If a model provides an incorrect sum, it is impossible to determine whether its visual understanding or its arithmetic capability is deficient.
>
> By stripping away these layers of arithmetic and logical complexity, ChartVRBench provides a clearer diagnostic signal of a model's core **visual** abilities, offering the community a targeted tool for identifying and resolving specific hallucinations related to geometric interpretation.

---

> ### Author Response · Authors · 2025-11-21
> **W4: Head-to-Head Comparisons and Effectiveness of VR-RFT**
>
> We appreciate the reviewer’s rigorous focus on experimental fairness and fully agree that isolating the algorithmic contribution from the strength of the backbone is crucial.
>
> To ensure a strictly fair comparison that isolates the training methodology, we deliberately included the state-of-the-art Bespoke-MiniChart-7B in our evaluation, as shown in Table 2. Since both Bespoke-MiniChart-7B and our ChartVR-7B are built on the exact same Qwen2.5-VL-7B backbone, we can fairly compare their performance to evaluate the training strategies. Despite sharing the same powerful foundation, our ChartVR-7B outperforms Bespoke-MiniChart-7B on ChartVRBench, achieving 72.20% compared to 68.16%, alongside CharXiv, ChartBench and ChartQAPro. This performance gap effectively showcases the superiority of our VR-RFT training strategy.
>
> Regarding the request to retrain previous methods, we respectfully clarify that applying our method to older backbones, such as the LLaVA-1.5 based ChartLlama, is technically infeasible. These older architectures typically rely on weaker vision encoders and language models which lack the native high-resolution dynamic encoding and modern long-horizon reasoning capabilities required to support the complex Chain-of-Thought reasoning our RL method relies on. Retraining them with our GRPO-based RL would likely result in model collapse.
>
> Furthermore, we have explicitly demonstrated our method's superiority over the traditional supervised paradigm in Section 5.3 and Table 3. In this controlled ablation, the CoT-SFT row corresponds to the traditional supervised fine-tuning technique applied to our data and backbone. The results show that the SFT method did not improve the model's performance and even caused a degradation to 54.06%, likely due to overfitting on reasoning formats without proper visual grounding. In contrast, our proposed Visual Reasoning Reinforcement Finetuning (VR-RFT) method achieved 72.20% on the same setup. This proves that simply having the data is insufficient and that our novel RL algorithm is essential to effectively elicit the true visual reasoning capability of MLLMs.

---

> ### Author Response · Authors · 2025-11-21
> **Q1: Evaluation of Intermediate Reasoning Steps**
>
> We sincerely thank the reviewer for this excellent question and for pointing out the importance of evaluating the intermediate reasoning steps.
>
> We would like to clarify that, consistent with the established standard in most chart-related benchmarks (such as ChartQA, ChartBench, and CharXiv) as well as broader reasoning benchmarks (like GSM8K or MATH), we primarily evaluate the accuracy of the final answer as **an effective proxy** for the model's targeted capability. The rationale behind this standard practice is that for complex tasks—particularly visual numerical estimation which requires high precision—it is statistically highly improbable for a model to arrive at a correct value within our strict 2% tolerance threshold without executing a correct underlying reasoning process. Therefore, the final answer serves as a robust and reliable indicator of the model's visual reasoning success.
>
> Nevertheless, to further validate that our VR-RFT training strategy genuinely enhances the quality of the overall reasoning process rather than just the final output, we conducted a rigorous post-hoc analysis using an "LLM-as-a-judge" methodology. We employed an advanced MLLM (*Qwen3-VL-32B-Instruct*) to evaluate the generated reasoning traces of both our model and the baseline on a 0-5 scale, assessing them for correctness, logic, and completeness.
>
> Our evaluation prompt explicitly state that:
> > - **5:** The reasoning is perfectly correct, logical, and easy to follow. All relevant visual elements are correctly identified, and the estimation/calculation is accurate.
> >- **4:** The reasoning is mostly correct and logical, but may contain a minor, inconsequential error or a slight imprecision in estimation that does not affect the overall conclusion.
> >- **3:** The overall reasoning direction is correct, but there is a noticeable error in visual identification, estimation, or a logical step. The process is still understandable.
> >- **2:** The reasoning contains significant flaws, such as incorrectly identifying a key visual element or making a major logical leap.
> >- **1:** The reasoning is completely incorrect, irrelevant to the question, or hallucinates information.
> >- **0:** The model failed to generate any discernible reasoning trace.
> >
>
>  As shown in the table below, ChartVR-7B achieves an average reasoning quality score of **4.09**, significantly outperforming the Qwen2.5-VL-7B baseline of **2.77**. Most notably, **55.6%** of ChartVR-7B's traces received a perfect score of 5 (indicating flawless logic), compared to only **30.5%** for the baseline, while the rate of complete reasoning failures (Score 0) dropped drastically from 34.0% to 6.9%. This confirms that our method effectively elicits high-quality, coherent visual reasoning traces.
>
> | **Model/Score** | 0 | 1 | 2 | 3 | 4 | 5 |
> | --- | --- | --- | --- | --- | --- | --- |
> | Qwen2.5-VL-7B | 34.0% | 0.2% | 3.6% | 9.2% | 22.5% | 30.5% |
> | **ChartVR-7B (Ours)** | 6.9% | 0.2% | 3.3% | 11.9% | 22.1% | 55.6% |

---

### Author Response · Authors · 2025-12-03
**Summary of Discussion**

We sincerely thank the reviewers and AC for their constructive feedback and rigorous scrutiny. As the discussion period concludes, we wish to summarize the core contributions of our work, clarify how we have addressed each reviewer’s concerns, and present additional experimental validations.

### Core Contributions

- **Isolation of Visual Reasoning via ChartVRBench (Reviewer kvg2, M2Y6, P44o):** We introduce ChartVRBench, a distinctive benchmark designed to isolate *visual reasoning* by focusing exclusively on numerical value estimation. By decoupling geometric perception from text recognition, we establish this estimation task as a foundational capability for chart understanding. The benchmark covers a broad spectrum of visual complexity, incorporating diverse chart types across both synthetic and real-world data, effectively serving as a precise diagnostic tool for current MLLM’s visual reasoning ability.
- **Enhanced Model Capabilities with VR-RFT (Reviewer zvmv, M2Y6):** We propose ChartVR, a series of MLLMs trained via a novel *Visual Reasoning Reinforcement Finetuning (VR-RFT)* strategy. ChartVR achieves superior performance on the challenging ChartVRBench, significantly outperforming general MLLMs and even powerful proprietary models.
- **Foundational Generalization (Reviewer kvg2, M2Y6):** Crucially, our VR-RFT strategy proves that reinforcing this *atomic visual capability* leads to generalized improvements across broad, complex chart tasks. We demonstrate that the visual reasoning ability cultivated by our method is not task-specific but foundational, achieving substantial performance gains across a diverse suite of public, multi-task chart understanding benchmarks.

---

> ### Author Response · Authors · 2025-12-03
> **Resolution of Concerns**
>
> **1. Benchmark Novelty & Necessity (Reviewer zvwv and M2Y6)**
>
> Reviewers questioned the distinction of ChartVRBench compared to existing benchmarks (e.g., ChartBench, EvoChart, PlotQA). We clarified that ChartVRBench is the first to strictly isolate visual reasoning from textual/logical reasoning, specifically by identifying numerical value estimation as the definitive task.
>
> - **Definition:** We differentiate Visual Reasoning (drawing inferences strictly from geometric properties, e.g., bar height relative to an axis) from Textual Reasoning (OCR extraction followed by logical deductions). Existing benchmarks conflate these capabilities, making it impossible to diagnose whether a failure stems from poor visual perception or flawed logic.
> - **Comparison:** We provided *a detailed feature-wise comparison table* demonstrating that ChartVRBench uniquely combines broad topic coverage, a strict focus on non-annotated charts, and the isolation of visual estimation. This fills a critical gap left by prior benchmarks that often rely on simplistic visual styles or values perfectly aligned with grid lines.
>
> **2. Data Integrity and Quality (Reviewer P44o and zvwv)**
>
> We addressed critical concerns regarding potential data contamination and benchmark quality with unambiguous confirmations and expanded analysis:
>
> - **Strict Train-Test Separation:** We clarified that ChartVRBench is *a strictly held-out test set*. The training data (SFT and RL sets) were generated using independent random seeds and distinct flows. There is *zero overlap* in images, questions, or answers between the training resources and the evaluation benchmark.
> - **Quality Assurance:** We detailed our rigorous "Human-in-the-loop" protocol, where annotators verified *100% of samples* to filter artifacts. Regarding complexity, we covered a wide range of chart categories—including composite charts—and employed strict prompt constraints to inject entropy (e.g., prohibiting integer alignment, enforcing non-monotonic trends). We have updated the manuscript with additional example images to visually demonstrate this structural diversity and difficulty.
>
> **3. Effectiveness of VR-RFT (Reviewer zvwv, M2Y6, and kvg2)**
>
> Reviewers requested fair head-to-head comparisons to isolate the efficacy of our VR-RFT strategy from the backbone strength.
>
> - **Controlled Comparison:** We introduced a comparison with Bespoke-MiniChart-7B and Chart-R1. Since Bespoke-MiniChart-7B, Chart-R1 and our ChartVR-7B all *share the exact same backbone (Qwen2.5-VL-7B)*, this comparison is architecturally fair. Empirically, ChartVR-7B outperforms all models on ChartVRBench and generalizes better across most public benchmarks, proving the superiority of our VR-RFT strategy over standard instruction tuning or other RL-based strategies.
> - **3B Model Performance:** We updated Table 2 to include ChartVR-3B, demonstrating that it outperforms similarly sized models (ChartGemma-3B, TinyChart-3B) across all benchmarks.
> - **Reasoning Step Evaluation:** We performed an LLM-as-a-judge evaluation, showing that our method reduces reasoning failures (Score 0) from 34.0% to 6.9% and significantly increases perfectly logical traces compared to the baseline.
>
> **4. Clarification on "Foundational" Skills and Distillation Limits (Reviewer P44o)**
>
> - **Scope of Generalization:** We clarified that our claim of "foundational" capability refers to the transfer of visual estimation skills across the chart domain. The success of our model on external, diverse benchmarks like CharXiv and ChartQAPro confirms this domain-generalization.
> - **Distillation Ceiling:** We addressed concerns about the student model being capped by the teacher's performance. We argue—supported by recent literature—that reasoning distillation often acts as a denoising process. The student model learns the robust underlying the reasoning logic while discarding the teacher's stochastic noise. This hypothesis is empirically supported by our results, where the distilled ChartVR model outperforms its teacher model, proving it has internalized the reasoning process rather than merely mimicking outputs.
>
> We believe these additional evaluations and clarifications firmly establish our ChartVRBench and ChartVR as a rigorous, novel, and highly effective contribution to the field of MLLM and chart understanding.
>
> Best regards,
>
> The Authors

---

> ### Author Response · Authors · 2025-12-03
> **List of Revisions**
>
> According to the reviewers' suggestions, we have submitted a revised version of the paper. The main revisions are as follows:
>
> - In **Section 3**, we have added a feature-wise comparison table (Table 1) to highlight the distinctions between ChartVRBench and existing benchmarks.
>
>     *Note: Due to this addition, references to Table 2 in our discussion correspond to Table 3 in the revised manuscript.*
>
> - In **Section 5.2**, we have updated Table 2 and Table 3 to include the experimental results of the Chart-R1 and the ChartVR-3B model, providing a fair head-to-head comparison to address the concerns regarding method effectiveness and model scaling.
> - In **Appendix B.1**, we have updated and clarified the SFT data curation process, explicitly confirming the strict separation between training resources and the evaluation benchmark, to address the data contamination concerns.
> - In **Appendix**, we have added Figure 5 displaying examples of complex synthetic charts (e.g., non-monotonic trends) within ChartVRBench, to address the concerns regarding data diversity and generative bias.

---

### Meta-Review · Area_Chair_cXxG · 2026-01-04

**Summary:**

This paper investigates whether current MLLMs truly perform visual reasoning when understanding charts, especially non-annotated ones. To this end, the authors introduce ChartVRBench, a benchmark intended to isolate visual numerical estimation from textual and logical reasoning, and propose ChartVR. The paper reports strong performance on the proposed benchmark and meaningful gains on several existing chart understanding benchmarks.

Reviewers generally agree that the problem formulation is well-motivated and that chart understanding without annotations represents a real and practically relevant failure mode of current MLLMs. The experimental results are solid, and the rebuttal substantially improved clarity around benchmark design, data separation, and experimental fairness. However, concerns remain regarding the degree of novelty, the distinctiveness of the benchmark relative to prior work, and the scope of the claimed generalization.

**Reviewer Concerns:**

Concerns addressed by the rebuttal:

- Train–test separation and data contamination
Initial concerns about potential overlap between training data and ChartVRBench were clearly addressed. The authors provided an unambiguous clarification that the benchmark is a strictly held-out test set, generated independently from the SFT and RL training data. This concern is considered resolved.

- Benchmark design rationale and comparison to existing datasets
The authors strengthened their argument that ChartVRBench differs from prior benchmarks by explicitly isolating visual numerical estimation and avoiding conflation with OCR-based or arithmetic reasoning. The added feature-wise comparison table and expanded discussion improved transparency and positioning.

- Effectiveness of the VR-RFT training strategy
Reviewers requested fairer comparisons to disentangle backbone strength from training methodology. The added controlled comparisons using identical backbones (e.g., Qwen2.5-VL-7B), ablations contrasting SFT vs. RL, and additional baselines (including Chart-R1) significantly strengthened the empirical case that VR-RFT provides tangible benefits.

Concerns that remain partially unresolved:

- Limited perceived novelty
Despite the clarifications, multiple reviewers remain unconvinced that the core contributions represent a sufficiently novel advance. Non-annotated chart understanding, numerical value estimation, and the use of RL or CoT-based supervision have all appeared in prior work. While ChartVRBench is more carefully scoped and diagnostic, its conceptual contribution is viewed by some reviewers as incremental rather than fundamentally new.

- Benchmark realism and scope
Although the authors addressed synthetic data bias through prompt constraints and human verification, questions remain about how closely the benchmark reflects the full diversity and complexity of real-world charts. Performance gaps on certain chart types (e.g., radar charts) further highlight limitations in generalization across fundamentally different visual structures.

- Strength of generalization claims
The paper frames visual numerical estimation as a “foundational” capability. While improvements on other chart benchmarks support in-domain transfer, reviewers noted that the evaluation remains restricted to chart-centric tasks. As a result, the broader framing of foundational reasoning may be somewhat overstated.

**Reviewer Scores:**

- Reviewer zvwv: Remains negative, though softened after rebuttal due to improved experimental fairness and clarification.

- Reviewer kvg2: Likely unchanged and mildly positive.

- Reviewer M2Y6: Borderline, acknowledging strong empirical results but viewing the contribution as incremental.

- Reviewer P44o: Remains negative, primarily due to concerns about novelty and the scope of generalization claims.

---

### Decision · Program_Chairs · 2026-01-26

Reject